# Controllable 3D Face Synthesis with Conditional Generative Occupancy Fields

**Keqiang Sun**[1]*,    **Shangzhe Wu**[2]*,    **Zhaoyang Huang**[1],
**Ning Zhang**[3],    **Quan Wang**[3],    **Hongsheng Li**[1,4]

[1]CUHK MMLab    [2]Oxford VGG    [3]SenseTime Research
[4]Centre for Perceptual and Interactive Intelligence Limited
`kqsun@link.cuhk.edu.hk, szwu@robots.ox.ac.uk, hsli@ee.cuhk.edu.hk`

## Abstract

Capitalizing on the recent advances in image generation models, existing controllable face image synthesis methods are able to generate high-fidelity images with some levels of controllability, e.g., controlling the shapes, expressions, textures, and poses of the generated face images. However, these methods focus on 2D image generative models, which are prone to producing inconsistent face images under large expression and pose changes. In this paper, we propose a new NeRF-based conditional 3D face synthesis framework, which enables 3D controllability over the generated face images by imposing explicit 3D conditions from 3D face priors. At its core is a conditional Generative Occupancy Field (cGOF) that effectively enforces the shape of the generated face to commit to a given 3D Morphable Model (3DMM) mesh. To achieve accurate control over fine-grained 3D face shapes of the synthesized image, we additionally incorporate a 3D landmark loss as well as a volume warping loss into our synthesis algorithm. Experiments validate the effectiveness of the proposed method, which can generate high-fidelity face images and shows more precise 3D controllability than state-of-the-art 2D-based controllable face synthesis methods. Find code and more demo at https://keqiangsun.github.io/projects/cgof.

## 1   Introduction

Recent success of Generative Adversarial Networks (GANs) [12] has led to tremendous progress in face image synthesis. State-of-the-art methods, such as StyleGAN [19, 20, 18], are capable of generating photo-realistic face images. Apart from photo-realism, being able to control the appearance of the generated images is also key in many real-world applications, such as face animation, reenactment, and free-viewpoint rendering. Early works on controllable face synthesis rely on external attribute annotations to learn an attribute-guided face image generation model [25, 10, 53]. However, these attributes, such as "big nose", "chubby" and "smiling" in CelebA dataset [24], can only provide abstract semantic-level guidance on the generation, and the generated faces often lack 3D geometric consistency. Moreover, it is often much harder to obtain low-level geometric annotations beyond semantic labels for direct 3D supervision.

Recently, researchers have attempted to incorporate 3D priors from parametric face models, such as 3D Morphable Models (3DMMs) [2, 36], into StyleGAN-based synthesis models, allowing for more precise 3D control over the generated images, including facial expressions and head poses [7, 50, 37]. Despite their impressive image quality, these models still tend to produce *3D inconsistent* faces under large expression and pose variations due to the lack of a 3D representation, as shown in Fig. 1.

---

*Equal Contribution

36th Conference on Neural Information Processing Systems (NeurIPS 2022).

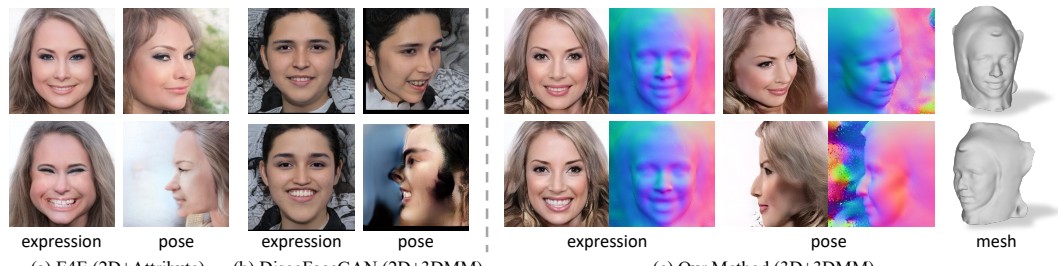

| expression | pose | expression | pose | expression | pose | mesh |
|---|---|---|---|---|---|---|

(a) E4E (2D+Attribute)  (b) DiscoFaceGAN (2D+3DMM)    (c) Our Method (3D+3DMM)

Figure 1: Existing controllable face synthesis methods, such as (a) E4E [51] and (b) DiscoFace-GAN [7], fail to preserve consistent geometry in the face images generated with large expression and pose variations, due to the lack of a 3D representation. (c) Our proposed *controllable 3D face synthesis* method leverages a NeRF-based 3D generative model conditioned on a prior 3DMM, which enables precise, disentangled 3D control over the generated face images.

With the advances in differentiable rendering and neural 3D representations, a recent line of work has explored photo-realistic 3D face generation using only 2D image collections as training data [47, 4, 43, 13, 35, 3, 55, 8]. Neural Radiance Fields (NeRFs) [29], in particular, have enabled 3D generative models, such as pi-GAN [4] and StyleNeRF [13], to synthesize high-fidelity, 3D consistent faces, by training with 2D images only. However, these models are purely generative and do not support precise 3D control over the generated faces, such as facial expressions.

In this work, we aim to connect these two groups of research—controllable face synthesis and 3D generative models—and present a NeRF-based 3D conditional face synthesis model that can generate high-fidelity face images with precise 3D control over the 3D shape, expression, and pose of the generated faces, by leveraging a parametric 3DMM face model. Imposing precise 3D conditions on an implicit neural radiance field is non-trivial. A naive baseline solution is to enforce that the input 3DMM parameters can be reproduced from the generated face image via a 3DMM reconstruction model, similar to [7]. Unfortunately, this 3DMM parameter reconstruction loss only provides *indirect* supervisory signals to the underlying NeRF volume, and is insufficient for achieving precise 3D control, as shown in Fig. 5.

We seek to impose explicit 3D condition *directly* on the NeRF volume. To this end, we propose a conditional Generative Occupancy Field (cGOF). It consists of a mesh-guided volume sampling procedure and a complimentary density regularizer that gradually concentrates the volume density around a given mesh. In order to achieve fine-grained 3D control, such as expression, we further introduce a 3D landmark loss and a volume warping loss.

The main contributions are as follows. We propose a novel controllable 3D face synthesis method that learns to generate high-fidelity face images with precise controllability over the 3D shape, expression, and poses, given only a collection of 2D images as training data. This is achieved by a novel conditional Generative Occupancy Field representation and a set of volumetric losses that effectively condition the generated NeRF volumes on a parametric 3D face model.

## 2   Related Work

**Controllable Generative Adversarial Networks.** Generative Adversarial Nets (GANs) [12] gained popularity over the last decade due to their remarkable image generation ability [40, 17, 19, 20]. Prior works have studied disentangled representation learning in generative models [14, 50, 49, 21, 44, 1, 30, 5, 6, 23]. Existing work on controllable facial image synthesis [38, 50, 49, 7, 37] relies on 2D image-based generative models, which does not guarantee 3D consistency in the generated images.

**3D-Aware GAN.** Another line of work in 3D-Aware generative models has looked into disentangling 3D geometric information from 2D images, and learning to generate various 3D structures, such as voxels [54, 58, 11, 30, 31, 26], meshes [47, 22], and Neural Radiance Fields (NeRFs) [4, 56, 13, 42, 34, 33], by training on image collections using differentiable rendering. The key idea is to use a discriminator that encourages images rendered from random viewpoints sampled from a prior pose distribution to be indistinguishable from real images. Among these works, pi-GAN has demonstrated high-fidelity 3D synthesis results adopting a NeRF representation. SofGAN [5]

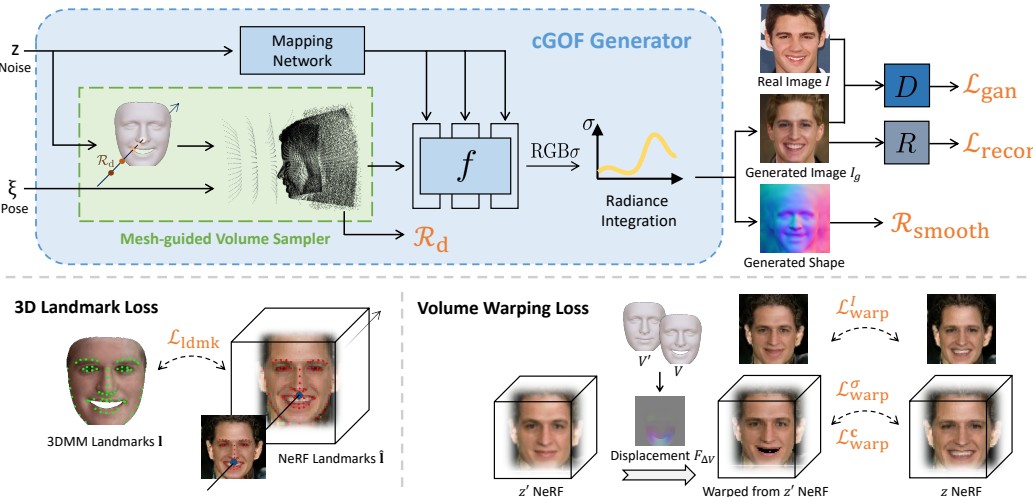

Figure 2: Method overview. *(Top)*: Conditional Generative Occupancy Field (cGOF) leverages a mesh-guided volume sampler which effectively conditions the generated NeRF on an input 3DMM mesh. It is trained in an adversarial learning framework using only single-view images. *(Bottom)*: The 3D landmark loss encourages the semantically important facial landmarks to follow the input mesh, and the volume warping loss enforces two NeRF volumes generated with different expression codes to be consistent through a warping field induced from the corresponding 3DMM meshes.

ensures viewpoint consistency via the proposed semantic occupancy field but requires 3D meshes with semantic annotations for pre-training. In this work, we build on top of pi-GAN and directly learn from unlabeled image collections with a prior 3DMM.

**3D Priors for Face Generation.** To learn controllable 3D face synthesis models, researchers have leveraged prior face models. 3D Morphable Models (3DMMs) [36, 2] is a widely-used parametric face model, which represents the human face as a set of PCA bases derived from 3D scans. Many previous works [7, 50, 5, 37] have proposed to incorporate 3D priors of 3DMM into generative models to enable controllable 3D face synthesis. In particular, DiscoFaceGAN [7], PIE [49] and StyleRig [50] leverage 3DMM to establish losses to ensure 3D consistency, but still use a 2D image-based StyleGAN generator. HeadNeRF [16] uses a 3D NeRF representation as well as 3DMM but is trained with a reconstruction loss using annotated multi-view datasets. Our method is trained on *unannotated single-view* images only, by directly imposing 3DMM conditions into 3D NeRF volume.

## 3 Method

Given a collection of single-view real-world face images $\mathcal{Y}$, where each face is assumed to appear only once without multiple views, the goal of this work is to learn a controllable 3D face synthesis model that can generate face images with desired 3D shape, expression, and pose. To achieve this, we propose a NeRF-based conditional 3D generative model $G$ and incorporate priors from a parametric 3D Morphable Model (3DMM) [36]. At the core of our method is a novel conditional Generative Occupancy Field (cGOF) as well as two auxiliary volumetric losses that enable effective 3D control based on the 3DMM, as illustrated in Figure 2.

### 3.1 Preliminaries on NeRF and pi-GAN

**Neural Radiance Field (NeRF).** NeRF [29] represents a 3D scene as a radiance field parametrized by a Multi-Layer Perceptron (MLP) that predicts a density value $\sigma \in \mathbb{R}$ and a radiance color $\mathbf{c} \in \mathbb{R}^3$ for every 3D location, given its $xyz$ coordinates and the viewing direction as input. To render a 2D image of the scene from an arbitrary camera viewpoint $\xi$, we cast rays through the pixels into the 3D volume, and evaluate the density values and radiance colors at a number of sampled points along each ray. Specifically, for each pixel, the camera ray $\mathbf{r}$ can be expressed as $\mathbf{r}(t) = \mathbf{o} + t\mathbf{d}$, where $\mathbf{o}$ denotes the camera origin, $\mathbf{d}$ the direction, and $t$ defines a sample point within near and far bounds $t_\mathrm{n}$ and $t_\mathrm{f}$. For each 3D sample point $\mathbf{x}_i$, we query the NeRF network $f$ to obtain its $\sigma_i$ and radiance color $\mathbf{c}_i$:

$(\sigma_i, \mathbf{c}_i) = f(\mathbf{x}_i, \mathbf{d})$. Using volume rendering [27, 29], the final color of the pixel is given by

$$\hat{C}(\mathbf{r}) = \sum_{i=1}^{N} T_i (1 - \exp(-\sigma_i \delta_i)) \mathbf{c}_i, \quad \text{where} \quad T_i = \exp(-\sum_{j=1}^{i-1} \sigma_j \delta_j), \tag{1}$$

and $\delta_i = t_{i+1} - t_i$ is the distance between adjacent samples. To optimize a NeRF on a 3D scene, the photometric loss is computed between the rendered pixels and the ground-truth images captured from a dense set of viewpoints.

**Generative Radiance Field.** Several follow-up works have extended this representation to a generative framework [4, 13, 8, 55]. Instead of using a multi-view reconstruction loss, they train a discriminator that encourages images rendered from randomly sampled viewpoints to be indistinguishable from real images, allowing the model to be trained with only single-view image collections.

Our method is built on top of pi-GAN [4], a generative radiance field model based on a SIREN-modulated NeRF representation [45]. During training, images are rendered from random viewpoints $\xi$ sampled from a predefined pose distribution $p_\xi$ (estimated from the training dataset). To model geometry and appearance variations, a random noise code $\mathbf{z}$ is sampled from a standard normal distribution and mapped to a number of frequencies $\gamma_i$ and phase shifts $\beta_i$ through a mapping network. These encodings are used to modulate the outputs of each layer in the NeRF MLP $f$ before the sinusoidal activations. The rendered images $I_g = G(\mathbf{z}, \xi)$ and the real images $I$ randomly sampled from the training dataset are then passed into the discriminator $D$, and the model is trained using a non-saturating GAN loss with R1 regularization following [28]:

$$\mathcal{L}(\theta) = \mathbb{E}_{\mathbf{z} \sim p_z, \xi \sim p_\xi}[f(D(G(\mathbf{z}, \xi)))] + \mathbb{E}_{I \sim p_\mathcal{D}}[f(-D(I)) + \lambda |\nabla D(I)|^2], \tag{2}$$

where $f(u) = -\log(1 + \exp(-u))$ and $\lambda$ is a hyperparameter balancing the regularization term.

### 3.2 Controllable 3D Face Synthesis

In order to learn a controllable 3D face synthesis model, we leverage priors of a 3DMM [36], which is a parametric morphable face model, where the shape, expression, texture, and other factors of a face are modeled by a set of PCA bases and coefficients $\tilde{\mathbf{z}} = (\tilde{\mathbf{z}}_{\text{shape}}, \tilde{\mathbf{z}}_{\text{exp}}, \tilde{\mathbf{z}}_{\text{tex}}, \tilde{\mathbf{z}}_{\text{else}})$ respectively. Our goal is to condition the generated face image $I_g$ on a set of 3DMM parameters $\tilde{\mathbf{z}}$ as well as a camera pose $\xi$, which specifies the desired configuration of the generated face image.

**Baseline.** To enforce such conditioning, following [7], we make use of an off-the-shelf pre-trained and fixed 3DMM reconstruction model $R$ that predicts a set of 3DMM parameters from a single 2D image [9]. This allows us to incorporate a 3DMM parameter reconstruction loss that ensures the generated face images commit to the input condition.

Specifically, we sample a random noise $\mathbf{z} \in \mathbb{R}^d$ from a standard normal distribution $p_{\mathbf{z}}$, which is later passed into the pi-GAN generator $G$ to produce a face image $I_g$. Unlike [7], which additionally trains a set of VAEs to map the noise $\mathbf{z}$ to meaningful 3DMM coefficients $\tilde{\mathbf{z}}$, we assume that the 3DMM coefficients of all the training images follow a normal distribution and simply standardize them to obtain the corresponding normalized noise code $\mathbf{z} = \tau(\tilde{\mathbf{z}}) = \tilde{L}^{-1}(\tilde{\mathbf{z}} - \tilde{\mu})$, where $\tilde{\mu}$ and $\tilde{L}$ are the mean and Cholesky decomposition component of covariance matrix of the 3DMM coefficients estimated from all the training images using the 3DMM reconstruction model $R$.

During training, we predict 3DMM parameters from the generated image using the reconstruction model $R$, normalize it and minimize a reconstruction loss between the predicted normalized parameters and the input noise:

$$\mathcal{L}_{\text{recon}} = \|\hat{\mathbf{z}} - \mathbf{z}\|_1, \quad \text{where} \quad \hat{\mathbf{z}} = \tau(R(G(\mathbf{z}, \xi))). \tag{3}$$

This results in a baseline model, where the conditioning is implicitly imposed through the 3DMM parameter reconstruction loss. Example results of this baseline model can be seen in Fig. 5.

**Challenges.** There are several issues with this baseline model. First, the effectiveness of the conditioning depends heavily on the accuracy and generalization performance of the 3DMM reconstruction model $R$. Fundamentally, these parameters only describe an extremely abstract, semantic representation of human faces, which is insufficient to govern the fine-grained geometric details in real faces precisely. Since our model generates a 3D representation for rendering, a more effective solution is

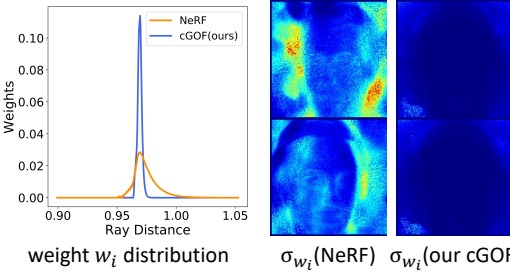 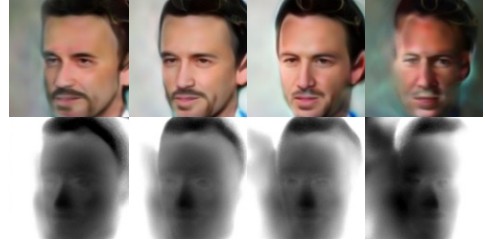

weight $w_i$ distribution   $\sigma_{w_i}$(NeRF) $\sigma_{w_i}$(our cGOF)   images/depths rendered from different poses

(a) Weight Distribution Comparison     (b) Shape-Texture Ambiguity w/o cGOF

Figure 3: Challenges in imposing 3D condition on NeRF. (a) Without the proposed cGOF (mesh-guided sampler and density regularizer), directly supervising the over-parameterized NeRF volume with a depth loss leads to ambiguous surfaces indicated by high variance of the rendering weights $\sigma_{w_i}$. (b) Rendering semi-transparent volumes leads to poor realism and multi-view consistency.

to impose the conditioning directly on the 3D volume, instead of just on the rendered 2D images. However, the radiance field representation is highly over-parametrized [57]. The color of a pixel is computed by integrating the radiance along the entire camera ray, whereas in many cases in reality, it should only reflect the color of the intersection point on an opaque surface. Incorporating losses directly on this over-parametrized 3D volume (*e.g.* a depth loss) tends to hamper its converging to an opaque surface, resulting in a semi-transparent volume that no longer guarantees multi-view consistency, as shown in Fig. 3.

### 3.3 Conditional Generative Occupancy Field

In order to impose precise 3D control on the NeRF volume, we first identify an effective way of enforcing the volume density to converge to the surface of human faces. Inspired by [55], we propose conditional Generative Occupancy Fields (cGOF), which consists of a mesh-guided volume sampler (MgS) and a distance-aware volume density regularizer.

The key idea is to use the conditioning 3DMM mesh to explicitly guide the volume sampling procedure. Given a random noise $\mathbf{z}$, we first obtain the corresponding 3DMM coefficients $\tilde{\mathbf{z}} = \tau^{-1}(\mathbf{z})$ by denormalization, which consists of $(\tilde{\mathbf{z}}_{\text{shape}}, \tilde{\mathbf{z}}_{\text{exp}}, \tilde{\mathbf{z}}_{\text{tex}}, \tilde{\mathbf{z}}_{\text{else}})$. With these coefficients, we can sample a 3D face mesh $M_{\text{in}}$ from the 3DMM $\Lambda$:

$$M_{\text{in}} = \Lambda(\mathbf{z}_{\text{shape}}, \mathbf{z}_{\text{exp}}) = M_{\text{mean}} + B_{\text{shape}} \cdot \tilde{\mathbf{z}}_{\text{shape}} + B_{\text{exp}} \cdot \tilde{\mathbf{z}}_{\text{exp}}, \tag{4}$$

where $M_{\text{mean}}$ is the mean shape, and $B_{\text{shape}}$ and $B_{\text{exp}}$ are the deformation bases for shape and expression in 3DMM respectively. During rendering, this mesh will serve as a 3D condition, allowing us to apply importance sampling around the surface region, and suppress the volume densities far away from the surface.

Specifically, for each pixel within the face region, we find its distance $t_{\text{m}}$ to the input 3DMM mesh $M_{\text{in}}$ along the ray direction by rendering a depth map, taking into account the perspective projection. This allows us to sample points along the ray $\mathbf{r}(t) = \mathbf{o} + t\mathbf{d}$ within a set of uniformly spaced intervals around the surface intersection point of the ray and the input mesh:

$$t_i \sim \mathcal{U}\left[t_{\text{m}} + (\frac{i-1}{N} - \frac{1}{2})\delta, t_{\text{m}} + (\frac{i}{N} - \frac{1}{2})\delta\right], \text{ where } i = 1, 2, \ldots, N_{\text{surf}}. \tag{5}$$

where $N_{\text{surf}}$ is the number of sampled points, and $\delta$ is a parameter controlling the "thickness" of the sampling region. We gradually shrink this margin from $0.5$ to $0.05$ of the volume range.

Furthermore, we suppress the volume density in the space far away from the given mesh using a distance-aware volume density regularizer. Specifically, we additionally sample a sparse set of points along each ray, and denote the absolute distances from these sampling points to the mesh surface as $d_i = |t_i - t_{\text{m}}|, i = 1, \ldots, N_{\text{vol}}$. The distance-aware volume density regularizer $R_{\text{d}}$ is defined as:

$$R_{\text{d}} = \sum_{i=0}^{N_{\text{vol}}} \sigma_i \cdot \left[\exp\left(\alpha \cdot \max(d_i - \delta/2, 0)\right) - 1\right], \tag{6}$$

where $\sigma_i$ is the volume density evaluated at those sample points, and $\alpha$ is an inverse temperature parameter (set to 20), which controls the strength of the penalty response to increased distance $\Delta d_i$.

Note that this penalty only applies to sample points beyond the surface sampling range $\frac{\delta}{2}$. The mesh-guided volume sampler together with the volume density regularizer effectively constrains the generated NeRF volume to follow closely the input 3DMM mesh, as shown in Tab. 2.

### 3.4 Auxiliary Losses for Fine-grained 3D Control

So far, the model is able to generate 3D faces following the overall shape of the conditioning 3DMM mesh. However, since 3DMM is only a low-dimensional PCA model of human faces without capturing the fine-grained shape and texture details of real faces, enforcing the generated 3D faces to fully commit to the coarse 3DMM meshes actually leads to over-smoothed shapes, and does not guarantee precise control over the geometric details, especially under large facial expression changes. To enable precise geometric control, we further introduce two auxiliary loss terms.

**3D Landmark Loss** encourages the semantically important facial landmarks to closely follow the input condition. Specifically, we identify a set of 3DMM vertices that correspond to $N_k = 68$ facial landmarks [41, 9]. Let $\{\mathbf{l}_k\}_{k=1}^{N_k}$ denote the 3D locations of these landmarks on the input 3DMM mesh. In order to find the same set of landmarks on the generated image, we extract 3D landmarks $\{\hat{\mathbf{l}}_k\}_{k=1}^{N_k}$ from the *estimated* 3DMM mesh and project them onto the 2D image to obtain the 2D projections. We then render the depth values of NeRF at these 2D locations using volume rendering and back-project them to 3D to obtain the 3D landmark locations of the generated volume $\{\hat{\mathbf{l}}_k'\}$. To avoid the impact of occlusion, we skip the facial contour landmarks (numbered from 1 to 17) when calculating loss. The landmark loss is defined among these three sets of 3D landmarks:

$$\mathcal{L}_{\text{ldmk}} = \sum_{k=1}^{N_k} \|\hat{\mathbf{l}}_k - \mathbf{l}_k\|_1 + \sum_{k=18}^{N_k} \|\hat{\mathbf{l}}_k' - \mathbf{l}_k\|_1. \tag{7}$$

Adopting this 3D landmark loss further encourages the reconstructed 3D faces to align with the input mesh at these semantically important landmark locations, which is particularly helpful in achieving precise expression control, as validated in Tab. 2.

**Volume Warping Loss** enforces that the NeRF volume generated with a different expression code should be consistent with the original volume warped by a warping field induced from the two 3DMM meshes. More concretely, let $M = \Lambda(\mathbf{z}_{\text{shape}}, \mathbf{z}_{\text{exp}})$ be a 3DMM mesh randomly sampled during training, $M' = \Lambda(\mathbf{z}_{\text{shape}}, \mathbf{z}_{\text{exp}}')$ be a mesh of the same face but with a different expression code $\mathbf{z}_{\text{exp}}'$, and $V$ and $V'$ be their vertices respectively. We first compute the vertex displacements denoted by $\Delta V = V' - V$, and render them into a 2D displacement map $F_{\Delta V} \in \mathbb{R}^{H \times W \times 3}$, where each pixel $F_{\Delta V}^{(u,v)}$ defines the 3D displacement of its intersection point on the mesh $M$.

For each pixel $(u, v)$ in the image $I_g$ generated from the first NeRF, we obtain the densities and radiance colors $(\sigma_i, \mathbf{c}_i) = f(\mathbf{x}_i, \mathbf{d}, \mathbf{z})$ of the $N_{\text{surf}}$ mesh-guided sample points $\mathbf{x}_i$ described above. The superscript $(u, v)$ is dropped for simplicity. We warp all the 3D sample points along each ray using the same 3D displacement value rendered at the pixel $F_{\Delta V}$ to obtain the warped 3D points $\mathbf{x}_i' = \mathbf{x}_i + F_{\Delta V}$ (assuming that all the points are already close to the mesh surface). We then query the second NeRF generated with $\mathbf{z}_{\text{exp}}'$ at these warped 3D locations $\mathbf{x}_i'$ to obtain another set of densities and radiance colors $(\sigma_i', \mathbf{c}_i') = f(\mathbf{x}_i', \mathbf{d}', \mathbf{z}')$, and encourage them to stay close to the original ones.

In addition, we also enforce the images rendered from the two radiance fields to be consistent. We integrate the radiance colors $\mathbf{c}_i'$ of the *warped* points with their densities $\sigma_i'$ using Eq. (1), and obtain an image $\hat{I}_g$ (with all pixels). This image $\hat{I}_g$ rendered from the second radiance field is encouraged to be identical to the original image $I_g$. The final warping loss is thus composed of three terms:

$$\mathcal{L}_{\text{warp}} = \beta_d \cdot \sum_{i}^{N_{\text{surf}}} \|\sigma_i' - \sigma_i\|_1 + \beta_c \cdot \sum_{i}^{N_{\text{surf}}} \|\mathbf{c}_i' - \mathbf{c}_i\|_1 + \beta_I \cdot \|\hat{I}_g - I_g\|_1, \tag{8}$$

where $\beta_d$, $\beta_c$ and $\beta_I$ are the balancing weights.

### 3.5 Neural Renderer and Overall Training Objective

Due to the computation- and memory-expensive volume rendering process, the quality of the generated images is bound to a small resolution ($64 \times 64$) in training. To improve the image quality without introducing too much computational overhead, we leverage an auxiliary neural renderer

| Method Name | Venue | $DS_s \uparrow$ | $DS_e \uparrow$ | $DS_p \uparrow$ | FID (512)$\downarrow$ |
|---|---|---|---|---|---|
| PIE [49] | TOG'20 | 1.66 | 15.24 | 2.65 | 59.63 |
| StyleRig [50] | CVPR'20 | 1.64 | 13.03 | 2.01 | 56.59 |
| DiscoFaceGAN [7] | CVPR'20 | 5.97 | 15.70 | 5.23 | 76.13 |
| E4E [51] | SIGGRAPH'21 | 1.91 | 8.66 | 7.08 | 96.40 |
| Gan-Control [44] | ICCV'21 | 7.07 | 7.51 | 9.33 | 83.59 |
| HeadNeRF [16] | CVPR'22 | 6.39 | 5.99 | 10.26 | 135.03 |
| Ours w/o Neural Renderer | - | **23.24** | **29.13** | **23.45** | 71.60 |
| Ours w/ Neural Renderer | - | 21.72 | 27.47 | 22.82 | **31.83** |

Table 1: DS and FID comparison with state-of-the-art controllable face synthesis methods.

introduced in [52] as a standalone module, which takes in the rendered coarse 2D image and produces a higher resolution version of it ($512 \times 512$). We take their pre-trained model on the FFHQ dataset and finetune it on the CelebA-HQ dataset. This neural renderer significantly improves the quality of the generated images without compromising much 3D controllability, as shown in Tab. 1.

Similar to RegNeRF [32], we further impose a smoothness regularizer $R_{smooth}$ on the generated NeRF volume by penalizing the L2 differences of neighboring pixels in both the depth maps and normal maps (referred to as $R_{smooth}^{depth}$ and $R_{smooth}^{norm}$ below). The final training objective is thus given by

$$\mathcal{L} = \lambda_{gan}\mathcal{L}_{gan} + \lambda_{recon}\mathcal{L}_{recon} + \lambda_d R_d + \lambda_{ldmk}\mathcal{L}_{ldmk} + \lambda_{warp}\mathcal{L}_{warp} + \lambda_{smooth}R_{smooth}, \quad (9)$$

where the $\lambda$'s specify the weights for each term.

## 4 Experiments

**Datasets.** We train our model on the CelebA [24] dataset, which consists of 200k celebrity face images with various poses, expressions, and lighting conditions. We use the cropped images and *do not make use of their facial attribute annotations*. The neural renderer module was originally trained on FFHQ dataset, which uses a slightly different cropping scheme. Therefore, we finetune their pre-trained model on the CelebA-HQ subset of CelebA containing 30k high-res images.

**Implementation Details.** We provide implementation details in the supplementary material, including network architectures and hyper-parameters. Here, we highlight a few important ones. We build our model on top of the implementation of pi-GAN [4], which uses a SIREN-based NeRF representation. We sample $N_{surf} = 12$ points around the 3DMM input mesh and $N_{vol} = 12$ coarse points for the density regularizer. The final model is trained for 72 hours on 8 GeForce GTX TITAN X GPUs.

**Metrics.** *(1) Disentanglement Score (DS)*. Following [7], we evaluate 3D controllability using a 3DMM Disentanglement Score. To measure the DS on one disentangled property $\mathbf{z}_i$ (*e.g.* shape, expression or pose), we randomly sample a set of $\mathbf{z}_i$ from a normal distribution while keeping other codes fixed $\{\mathbf{z}_j\}, j \neq i$, and render images. We re-estimate the 3DMM parameters $\hat{\mathbf{z}}_i$ and $\{\hat{\mathbf{z}}_j\}$ from these rendered images and compute $DS_i = \prod_{j,j \neq i} \sigma_{\hat{\mathbf{z}}_i}/\sigma_{\hat{\mathbf{z}}_j}$, where $\sigma_{\hat{\mathbf{z}}_i}$ and $\sigma_{\hat{\mathbf{z}}_j}$ denote variance of the reconstructed parameters of the measured property $\hat{\mathbf{z}}_i$ and the rest $\hat{\mathbf{z}}_j$, respectively. A higher DS score indicates that the model achieves more precise controllability on that property while keeping the others unchanged. *(2) Chamfer Distance (CD)*. We report the bi-directional Chamfer Distance between the generated face surface and the input 3DMM to measure how closely the generated 3D face follows the condition. *(3) Landmark Distance (LD)*. To evaluate the fine-grained controllability of our model, we detect the 3D landmarks in the generated images using [9] and compute the Euclidean distance between the detected landmarks and the ones extracted from the input 3DMM vertices. *(4) Landmark Correlation (LC)*. We measure the precision of expression control, by computing the 3D landmark displacements of two faces generated with a pair of random expression codes, and reporting the correlation score between these displacements and those obtained from the two 3DMM meshes. *(5) Frechet Inception Distance (FID)*. The quality of the generated face images is measured using Frechet Inception Distance [15] with an ImageNet-pretrained Inception-V3 [48] feature extractor. We align images with 68 landmarks [41, 46, 39] to avoid domain shift.

### 4.1 Qualitative Evaluation

Fig. 4 shows some example images generated by our proposed cGOF model. In particular, we visualize the generated faces by varying only one of the controlled properties at a time, including

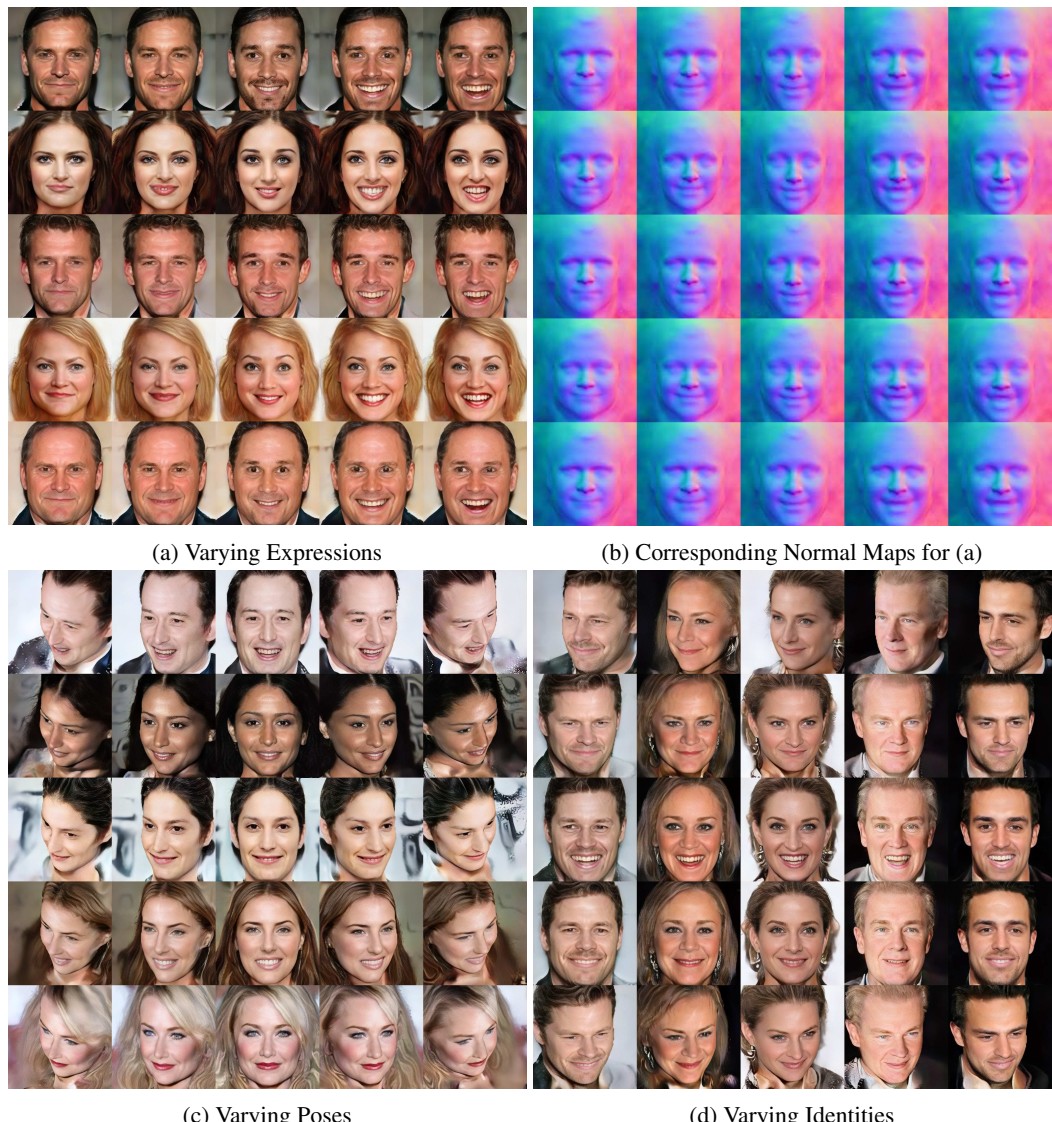

|  |  |
|---|---|
| (a) Varying Expressions | (b) Corresponding Normal Maps for (a) |

|  |  |
|---|---|
| (c) Varying Poses | (d) Varying Identities |

Figure 4: Face images generated by the proposed cGOF. The variations in the expression, pose and identity of the faces are highly disentangled and can be precisely controlled.

facial expression, head pose, and identity–combining shape and texture as in [7]. Our model is able to generate high-fidelity 3D faces with precise 3D controllability in a highly disentangled manner, even in the presence of large expression and pose variations. In contrast, existing methods tend to produce inconsistent face images as shown in Fig. 1. Please refer to the supplementary materials for more qualitative results and side-by-side comparisons against state-of-the-art methods.

## 4.2 Quantitative Evaluation

We compare our method to several state-of-the-art methods. E4E [51] and GAN-Control [44] are purely 2D controllable face image synthesis methods, which are based on facial attribute annotations. DiscoFaceGAN [7], PIE [49] and StyleRig [50] leverage 3DMM priors but still use a 2D image-based StyleGAN generator. HeadNeRF [16] uses a 3D NeRF representation as well as 3DMM but is trained with a reconstruction loss using annotated datasets, whereas our method is a 3D generative model trained on *unannotated single-view* images only.

We compare the controllability accuracy using the Disentanglement Score (see Col 3 to 5 of Tab. 1). Our method achieves significantly better disentanglement and more precise control in terms of shapes,

| Index | Loss | CD ↓ | LD ↓ | LC ↑ | $DS_s$ ↑ | $DS_e$ ↑ | $DS_p$ ↑ | FID (128) ↓ |
|---|---|---|---|---|---|---|---|---|
| 1 | $\mathcal{L}_{gan}$ | 1.09 | 5.04 | 2.04 | 2.13 | 2.54 | 7.16 | **18.76** |
| 2 | $+ \mathcal{L}_{recon}$(baseline) | 0.87 | 3.85 | 26.15 | 3.56 | 5.03 | 11.00 | 21.97 |
| 3 | $(+ \mathcal{L}_{depth})$* | 1.65 | 6.16 | 0.06 | 0.93 | 1.34 | 1.09 | 71.67 |
| 4 | + MgS | 0.29 | 3.98 | 27.45 | 3.55 | 4.90 | 9.48 | 38.91 |
| 5 | $+ \mathcal{R}_d$ | 0.31 | 3.51 | 51.74 | 3.56 | 5.31 | 15.94 | 29.62 |
| 6 | $+ \mathcal{R}_{smooth}^{norm}$ | 0.27 | 4.72 | 30.25 | 3.18 | 4.43 | 16.26 | 31.63 |
| 7 | $+ \mathcal{L}_{ldmk}$ | 0.39 | 1.86 | 84.43 | 16.54 | 16.65 | 21.24 | 56.90 |
| 8 | $+ \mathcal{L}_{warp}$ | **0.26** | 1.44 | 89.91 | 20.47 | 22.04 | 22.91 | 47.18 |
| 9 | $+ \mathcal{R}_{smooth}^{depth}$ | 0.27 | **1.26** | **92.88** | **23.24** | **29.13** | **23.45** | 26.64 |

Table 2: Ablation Study. Each row adds an additional loss term into the objective function upon all the loss terms above it. The last row is our final model. *Note that we do not use the depth loss in the final model, which compares the depths rendered from NeRF and 3DMM.

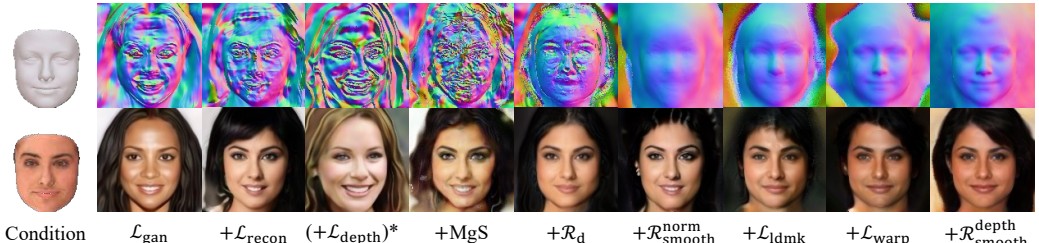

Condition    $\mathcal{L}_{gan}$    $+\mathcal{L}_{recon}$   $(+\mathcal{L}_{depth})$*   +MgS    $+\mathcal{R}_d$   $+\mathcal{R}_{smooth}^{norm}$   $+\mathcal{L}_{ldmk}$   $+\mathcal{L}_{warp}$   $+\mathcal{R}_{smooth}^{depth}$

Figure 5: Ablation Study. We analyze the effects of the individual components and loss functions. With the components gradually enabled, the model is able to generate 3D realistic faces precisely following the condition 3DMM. Refer to Sec. 4.3 for more details. *Note that we do not use the depth loss in the final model, which compares the depths rendered from NeRF and 3DMM.

expressions, and poses, compared to existing methods. Note that since PIE and StyleRig do not have open-sourced code, we compute the score on the 167 generated images provided on the PIE project page[2]. We also compare the FID scores in Col 6 of Tab. 1. Our model achieves plausible image quality without the Neural Renderer, and outperforms the existing methods with the Neural Renderer.

### 4.3 Ablation Study

We conduct ablation studies to validate the effectiveness of each component in our model by adding them one by one, and evaluating the generated images with the four metrics specified above. Tab. 2 and Fig. 5 illustrate the studies and show that all the components improve the precision of 3D controllable image generation. In particular, the mesh-guided volume sampler (MgS) significantly improves the effectiveness of the 3DMM condition (denoted as '+ MgS' in Tab. 2). The 3D landmark loss $\mathcal{L}_{ldmk}$ (denoted as '+ $\mathcal{L}_{ldmk}$') and the volume warping loss $\mathcal{L}_{warp}$ (denoted as '+ $\mathcal{L}_{warp}$') further enhance the control over fine-grained geometric details. We also report FIDs of the outputs of the cGOF with the image resolution of $128 \times 128$. Overall, although the geometric regularizers lead to slightly degraded image quality (indicated by increased FIDs), they improve the 3D controllability by large margins (indicated by the improvement in other metrics).

## 5 Conclusions

We have presented a controllable 3D face synthesis method that can generate high-fidelity face images based on a conditional 3DMM mesh. We propose a novel conditional Generative Occupancy Field representation and a set of 3D losses that effectively impose 3D conditions directly on the generated NeRF volume, which enables much more precise 3D controllability over 3D shape, expression, and pose of the generated faces than state-of-the-art 2D counterparts. **Limitations.** The proposed method still lacks precise texture and illumination control over the generated face images. It also relies heavily on the prior 3DMM and does not generate high-fidelity shape details, such as wrinkles and hair. **Societal Impacts.** The proposed work might be used to generate fake facial images for imposture or

---

[2]https://vcai.mpi-inf.mpg.de/projects/PIE/

to create synthesized facial images to hide the true identities of cybercriminals. Defensive algorithms for recognizing such synthesized facial images should be accordingly developed.

# 6 Acknowledgements

This work is supported in part by Centre for Perceptual and Interactive Intelligence Limited, in part by the General Research Fund through the Research Grants Council of Hong Kong under Grants (Nos. 14204021, 14207319).

We would like to thank Yu Deng for providing the evaluation code of the Disentanglement Score, and Jianzhu Guo for providing the pre-trained face reconstruction model. We are also indebted to thank Xingang Pan, Han Zhou, KwanYee Lin, Jingtan Piao, and Hang Zhou for their insightful discussions.

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
