# Controllable 3D Face Synthesis with Conditional Generative Occupancy Fields
## –Supplementary Material–

**Keqiang Sun**[1]*    **Shangzhe Wu**[2]*,    **Zhaoyang Huang**[1],
**Ning Zhang**[3],    **Quan Wang**[3],    **Hongsheng Li**[1,4]
[1]CUHK MMLab    [2]Oxford VGG    [3]SenseTime Research
[4]Centre for Perceptual and Interactive Intelligence Limited
kqsun@link.cuhk.edu.hk, szwu@robots.ox.ac.uk, hsli@ee.cuhk.edu.hk

## 1   Implementation Details

**Network Architectures.**   This project is based on the released code of pi-GAN [2] and Deep3DFaceRecon [4]. We use the same architectures for the generator and discriminator, as well as the main training process from the pi-GAN. To implement our proposed conditional Generative Occupancy Field (cGOF), we integrate a deep 3D face reconstruction model Deep3DFaceRecon [4] to reconstruct 3DMM parameters from the generated images for the 3DMM reconstruction parameter loss $\mathcal{L}_{\text{recon}}$. We follow [4] to adopt the popular 2009 Basel Face Model [9] for shape and texture bases, and use the expression bases of [6], built from FaceWarehouse [1]. The expression components are the PCA model of the offsets between the expression meshes and the neutral meshes of individual persons.

**Coordinate System Alignment.**   We align the 3DMM meshes to pi-GAN to impose the 3D losses. We first use the pre-trained pi-GAN to generate multi-view images of various face instances in the canonical viewpoint and use the Deep3DFaceRecon to reconstruct 3DMM meshes $M$ for the generated faces, on which we define a set of landmarks $\mathbf{l}_{\text{3D}}$. Let $K_{\text{recon}}$ and $E_{\text{recon}}$ be the intrinsic and extrinsic matrices for 3DMM, and $K_{\text{pigan}}$ and $E_{\text{pigan}}$ the counterparts for pi-GAN respectively. Let $T_{\text{R2P}}$ be the transformation matrix from 3DMM to pi-GAN that we would like to estimate. We directly optimize this transformation matrix by minimizing the difference of 2D projections of a set of landmarks obtained from the original mesh $M$ and the transformed mesh $M' = T_{\text{R2P}} \cdot M$, denoted as $\mathbf{l}_{\text{2D}}$ and $\mathbf{l}'_{\text{2D}}$:

$$T_{\text{R2P}}^* = \underset{T_{\text{R2P}}}{\arg\min} \|\mathbf{l}_{\text{2D}} - \mathbf{l}'_{\text{2D}}\|_1,$$
$$\text{where} \quad \mathbf{l}_{\text{2D}} = K_{\text{recon}} \cdot E_{\text{recon}} \cdot \mathbf{l}_{\text{3D}},$$
$$\mathbf{l}'_{\text{2D}} = K_{\text{pigan}} \cdot E_{\text{pigan}} \cdot T_{\text{R2P}} \cdot \mathbf{l}_{\text{3D}}. \tag{1}$$

**Background Modeling.**   We model the background by setting the weight of the last sample point along each ray to be $w_N = 1 - \sum_{i=1}^{N-1} w_i$, where $w_i = T_i \cdot (1 - \exp(-\sigma_i \delta_i))$ is weight of the rest of the sample points along the ray, including both the mesh-guided samples and the volume samples.

**Hyper-Parameters and Volume Sampling.**   Tab. 1 summarizes all hyper-parameters. We sample 12 coarse points evenly in the volume to and obtain $N_{\text{vol}} = 12$ fine points as introduced in [8]. Only $N_{\text{vol}} = 12$ fine points and the $N_{\text{surf}} = 12$ points sampled around the 3DMM input mesh are used for rendering and optimization with gradient backpropagation. The final model is trained for 72 hours on 8 GeForce GTX TITAN X GPUs.

---

*Equal Contribution

| Parameter | Value/Range |
|---|---|
| Optimizer | Adam |
| Generator learning rate | $6 \times 10^{-5}$ |
| Discriminator learning rate | $2 \times 10^{-4}$ |
| Number of iterations | $80,000$ |
| Batch size | 64 |
| $N_{\text{surf}}$ | 12 |
| $N_{\text{vol}}$ | 12 |
| training image size | 64 |
| Loss weight $\lambda_{\text{gan}}$ | 1 |
| Loss weight $\lambda_{\text{recon}}$ | 4 |
| Loss weight $\lambda_{\text{d}}$ | 10000 |
| Loss weight $\lambda_{\text{ldmk}}$ | 10 |
| Loss weight $\lambda_{\text{warp}}$ | 10 |
| Loss weight $\lambda_{\text{smooth}}^{\text{norm}}$ | 1000 |
| Loss weight $\lambda_{\text{smooth}}^{\text{depth}}$ | 500 |
| $\mathbf{z}$ | $\mathcal{N}^{411}(0,1)$ |
| Ray length | $(0.88, 1.12)$ |
| Yaw | $\mathcal{N}(0, 17.19°)$ |
| Pitch | $\mathcal{N}(0, 8.88°)$ |
| Field of view (FOV) | $12°$ |

Table 1: Training details and hyper-parameter settings.

To determine loss weights $\lambda$s, we start from the original Pi-GAN and add the proposed components one-by-one, as in **??** in the main paper. For each component, we first initialize the hyperparameter lambda as 1, and empirically find a reasonable value before taking a fine-grained search.

## 2 Additional Quantitative Results

### 2.1 Further Comparison with DiscoFaceGAN

In **??** in the main paper, we have compared with DiscoFaceGAN [3] in terms of Disentangle Scores. To directly compare the 3DMM conditioning methods proposed in our paper with those in the DiscoFaceGAN [3], we introduce the conditioning methods of DiscoFaceGAN, *i.e.* imitative loss, and contrastive loss, into Pi-GAN to build the "Pi-GAN + DiscoFaceGAN" model.

Comparison results are shown in the Tab. 2. Row 1 shows the Disentangle Scores of the officially released DiscoFaceGAN model concerning shapes, expressions, and poses. Row 2 corresponds to the constructed "Pi-GAN + DiscoFaceGAN" method. Comparing Row 1 and 2, we find the Pi-GAN backbone enhances the disentanglement of head pose and face shape while harming the expression disentanglement. Row 3 indicates the performance of our method, which outperforms the "Pi-GAN + DiscoFaceGAN" model in all metrics by a large margin, demonstrating the efficiency of the proposed conditioning method against the DiscoFaceGAN [3].

### 2.2 User Study

We conduct a user study to add to the comparison. We follow the experiment setting of **??** in the main paper. Specifically, we provide the control results of five methods, and ask a total of 21 users to rank the results according to their image quality and the controlling effects.

Tab. 3 summarizes the results, where the "Average Ranking / Average Score / Ranking 1st Ratio" are reported for each method with respect to concerning factors, *i.e.* identity, expression, pose. Our model achieves the highest average ranking and scores for all aspects, indicating our model produces more perceptually compelling results and achieves better 3D controllability than other counterparts.

| Index | Loss | CD $\downarrow$ | LD $\downarrow$ | LC $\uparrow$ | $DS_\text{s} \uparrow$ | $DS_\text{e} \uparrow$ | $DS_\text{p} \uparrow$ |
|---|---|---|---|---|---|---|---|
| 1 | Pi-GAN | 1.09 | 5.04 | 2.04 | 2.13 | 2.54 | 7.16 |
| 2 | DiscoFaceGAN [3] | - | - | - | 5.97 | 15.70 | 5.23 |
| 3 | Pi-GAN + DiscoFaceGAN | 1.32 | 2.40 | 32.55 | 6.16 | 7.63 | 12.13 |
| 4 | Ours | **0.27** | **1.26** | **92.88** | **23.24** | **29.13** | **23.45** |

Table 2: Comparison with "Pi-GAN + DiscoFaceGAN".

| | DiscoFaceGAN [3] | GAN-Control [10] | Pi-GAN [2] + $\mathcal{L}_\text{recon}$ | HeadNeRF [7] | cGOF (Ours) |
|---|---|---|---|---|---|
| Id | 3.1 / 59.1 / 9.52% | 2.6 / 68.6 / 0.00% | 3.3 / 53.3 / 9.52% | 4.7 / 26.7 / 0.0% | **1.4 / 92.4 / 81.0%** |
| Exp | 2.3 / 74.3 / 19.1% | 3.8 / 43.8 / 9.52% | 3.2 / 56.2 / 0.00% | 4.3 / 34.3 / 0.0% | **1.4 / 91.4 / 71.4%** |
| Pose | 2.6 / 68.6 / 14.3% | 3.7 / 45.7 / 0.00% | 3.8 / 43.8 / 4.76% | 3.7 / 46.7 / 0.0% | **1.2 / 95.2 / 81.0%** |
| IQ | 2.1 / 78.1 / 9.52% | 2.9 / 62.9 / 4.76% | 3.9 / 42.9 / 0.00% | 5.0 / 20.0 / 0.0% | **1.2 / 96.2 / 85.7%** |

Table 3: User Study on the Disentangle Performance and Image Quality. Each cell contains "Average Ranking / Average Score / Ranking 1st Ratio". Id: Identity, Exp: Expression, IQ: Image Quality.

# 3 Additional Qualitative Results

## 3.1 Additional Results with Pose Variations

We present large pose results for GAN-Control [10] in Fig. 1, DiscoFaceGAN [3] in Fig. 2, Head-NeRF [7] in Fig. 3 and pi-GAN [2] in Fig. 4. Previous methods fail to generate plausible face images when the camera pose gets larger (*e.g.* $> 60°$), including NeRF-based methods (HeadNeRF and pi-GAN). Nevertheless, as shown in Fig. 5, our method produces plausible *3D consistent* face images even in extremely large poses. Note that, to get rid of the view-dependent effect, we follow [5, 7] and remove the dependence of the radiance colors on the viewing direction by setting the view direction to a constant $(0, 0, -1)$ when evaluating the radiance colors of the sample points.

## 3.2 Additional Results with Expression Variations

We present more results on the expression control, comparing our method against two state-of-the-art controllable face synthesis methods, one attribute-guided GAN-Control [10], and the other 3DMM-guided DiscoFaceGAN [3]. Figs. 6 to 8 show the generated faces using GAN-Control, DiscoFaceGAN and our method respectively. For each figure, the first column shows a reference image, columns 2 to 5 show images generated with mild expressions, and columns 6 to 9 show images generated with wilder expressions. Each row corresponds to the same person, and each column corresponds to the same expression.

In Fig. 6, we can see that GAN-Control fails to preserve the identity as well as other factors of the face image (*e.g.* background) when changing only the facial expression code. A few examples are highlighted in red. Moreover, we observe that with the original range of the expression parameters, the model results in only a small variation of expressions, whereas with increased perturbations, it leads to much more significant shape and identity inconsistencies. We also notice the "smiling" attribute tends to strongly correlate with the "female" and "long-hair" attributes.

In Fig. 7, we can see that the DiscoFaceGAN fails to impose consistent expression control over the faces. The blue boxes highlight a few examples, where the same expression code produces different expressions in different faces. Moreover, the images generated with wild expressions may appear unnatural, such as 'exp 5' in Fig. 7.

In Fig. 8, we show that our model generates compelling photo-realistic face images with highly consistent, precise expression control. In each row, only expressions change while other properties remain unchanged, such as identity (shape and texture), hair, and background. In each column, all instances follow the same expression.

In addition, in Fig. 9, we present examples of images generated by our model with *out-of-distribution* expressions, such as frowning, pouting, curling lips, smirking *etc.* Despite images with such expressions hardly existing in the training data, our model is still able to generate highly plausible images.

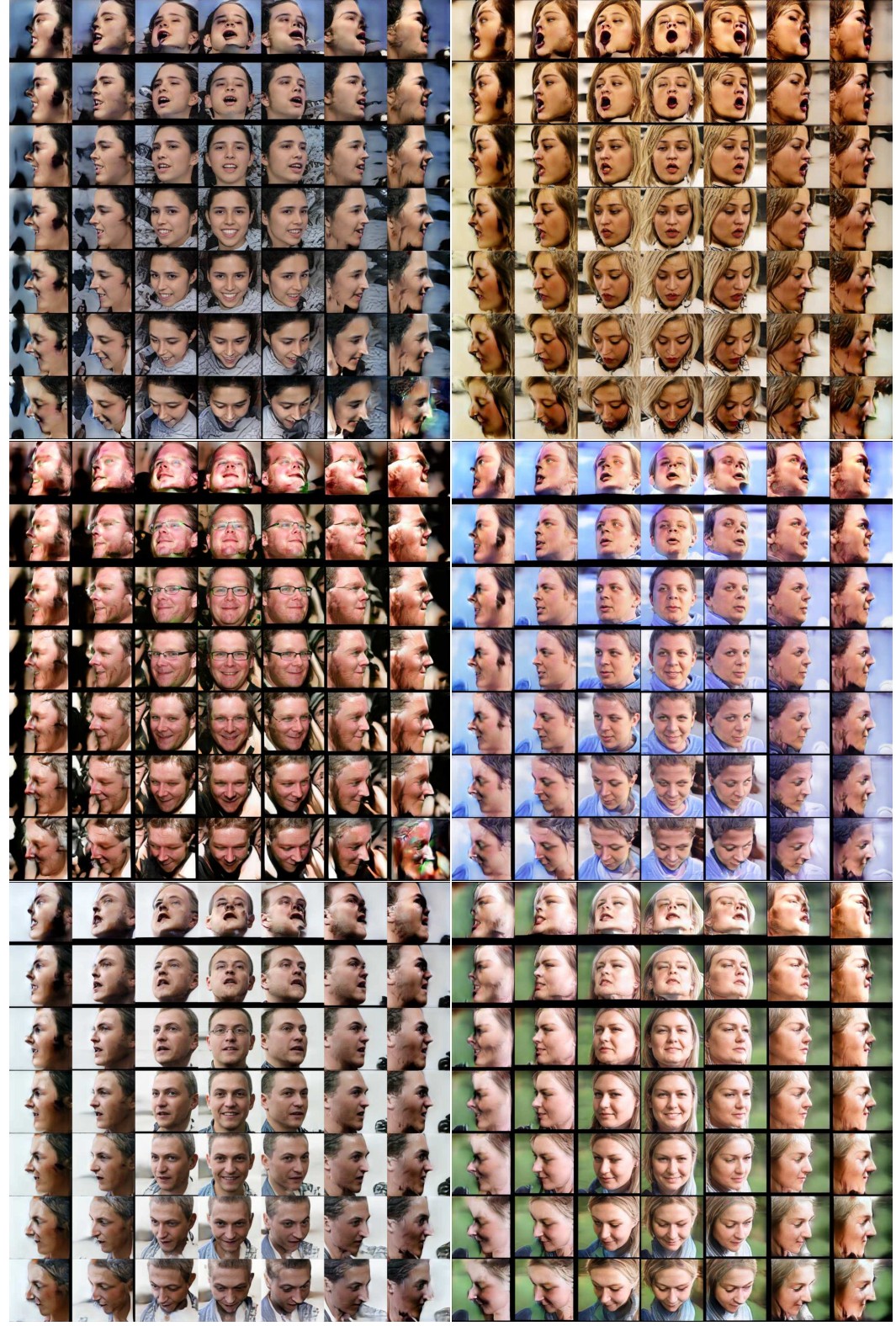

Figure 1: Images generated by DiscoFaceGAN [3] with different *poses*.

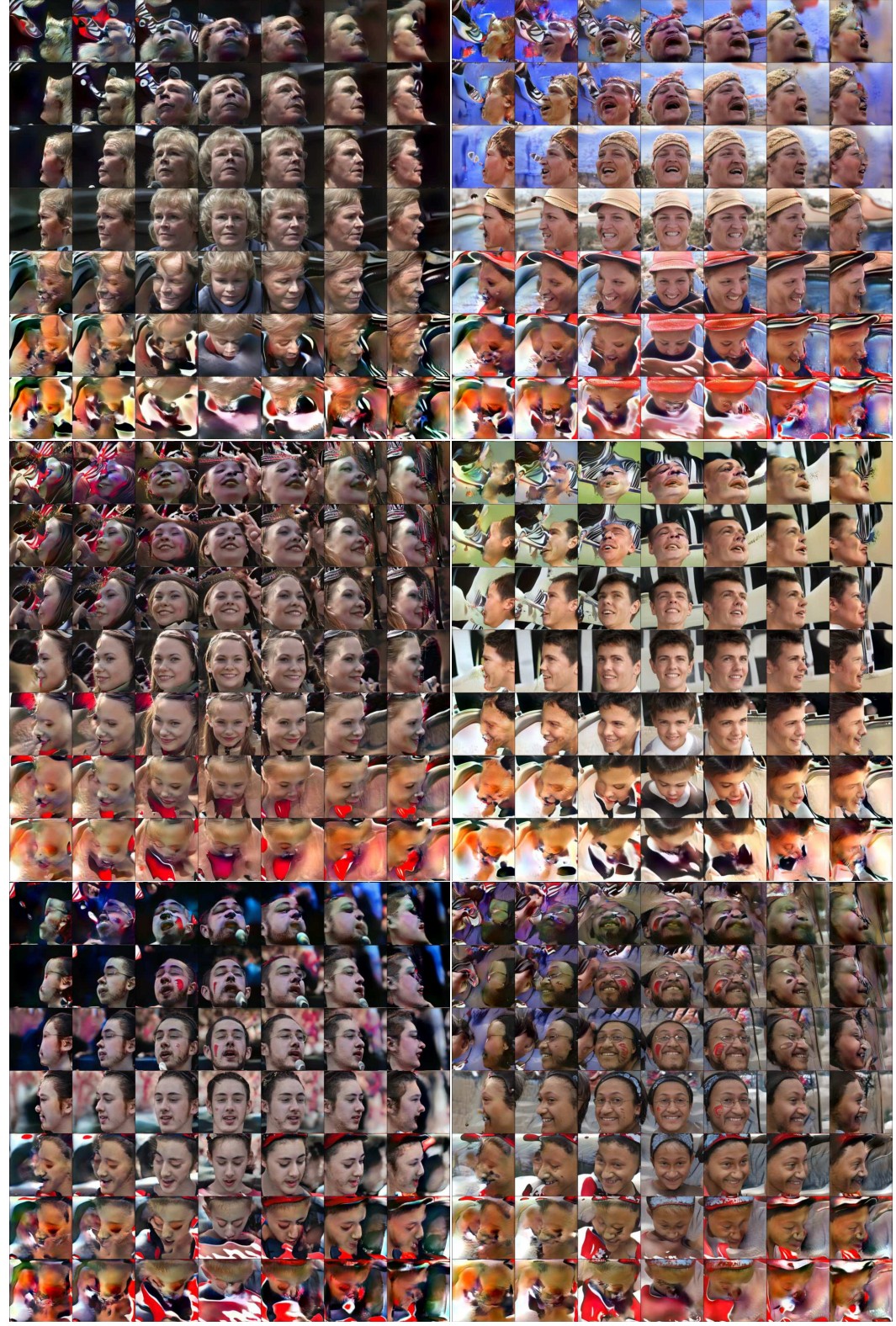

Figure 2: Images generated by GAN-Control [10] with different *poses*.

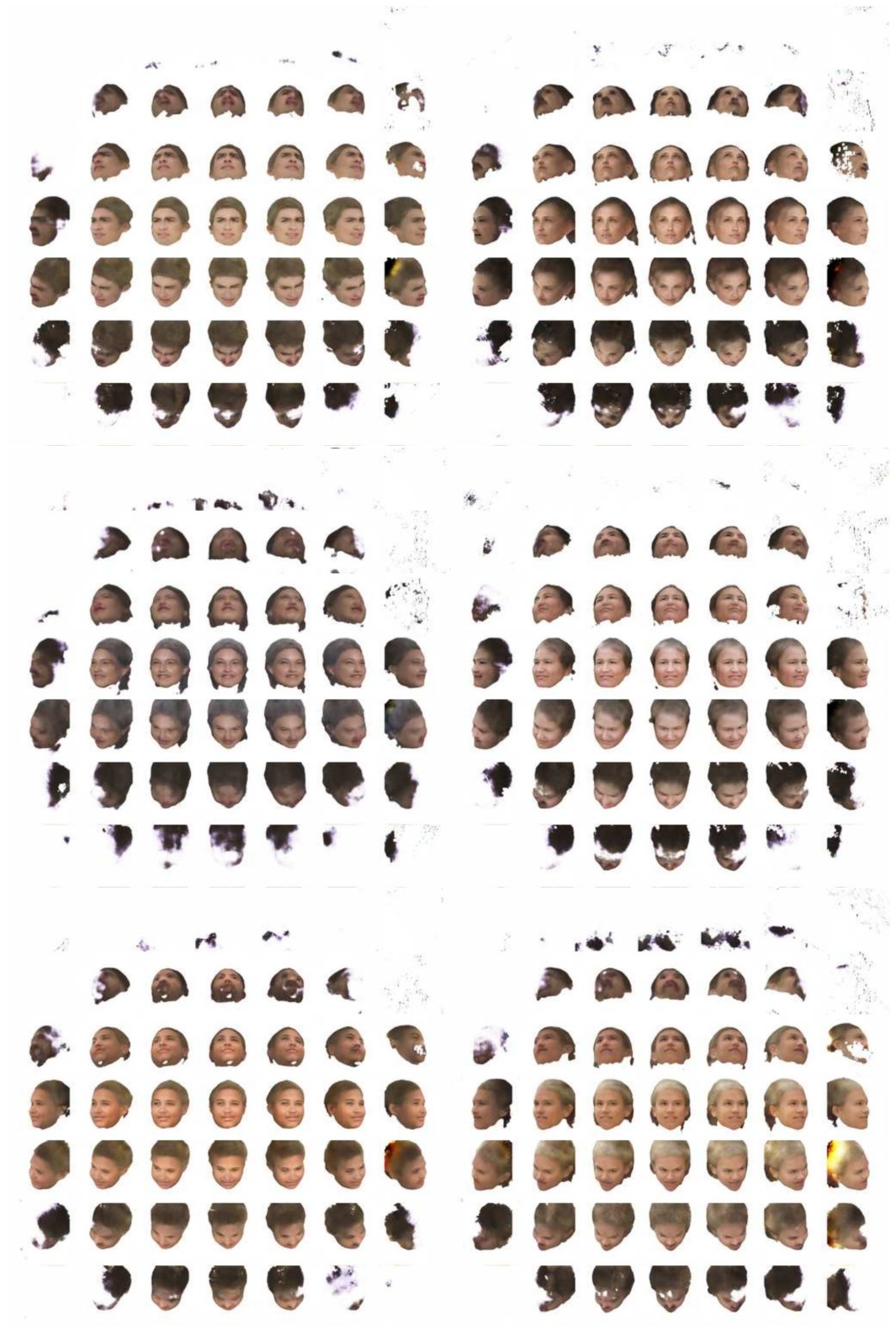

Figure 3: Images generated by HeadNeRF [7] with different *poses*.

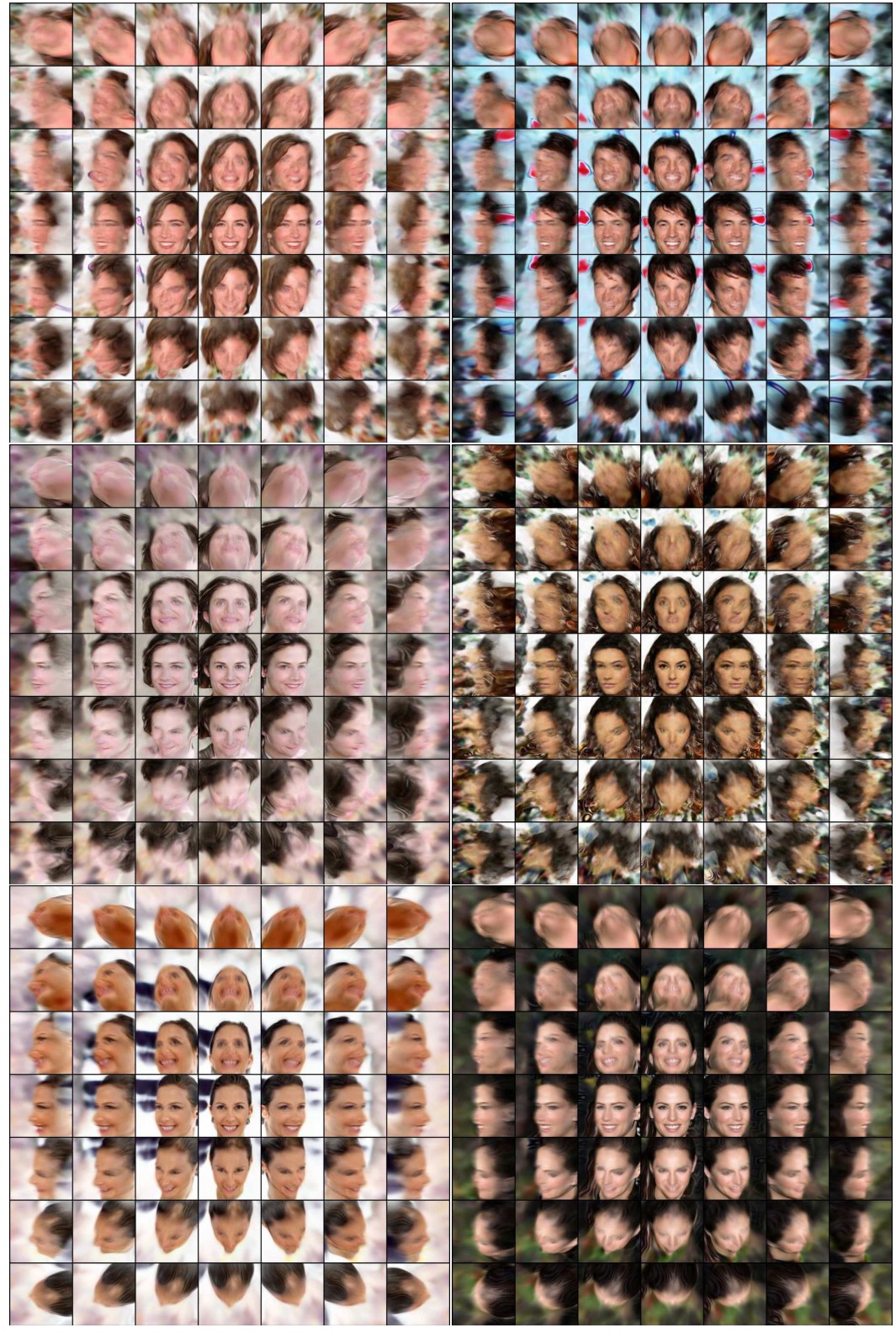

Figure 4: Images generated by the original Pi-GAN [2] with different *poses*.

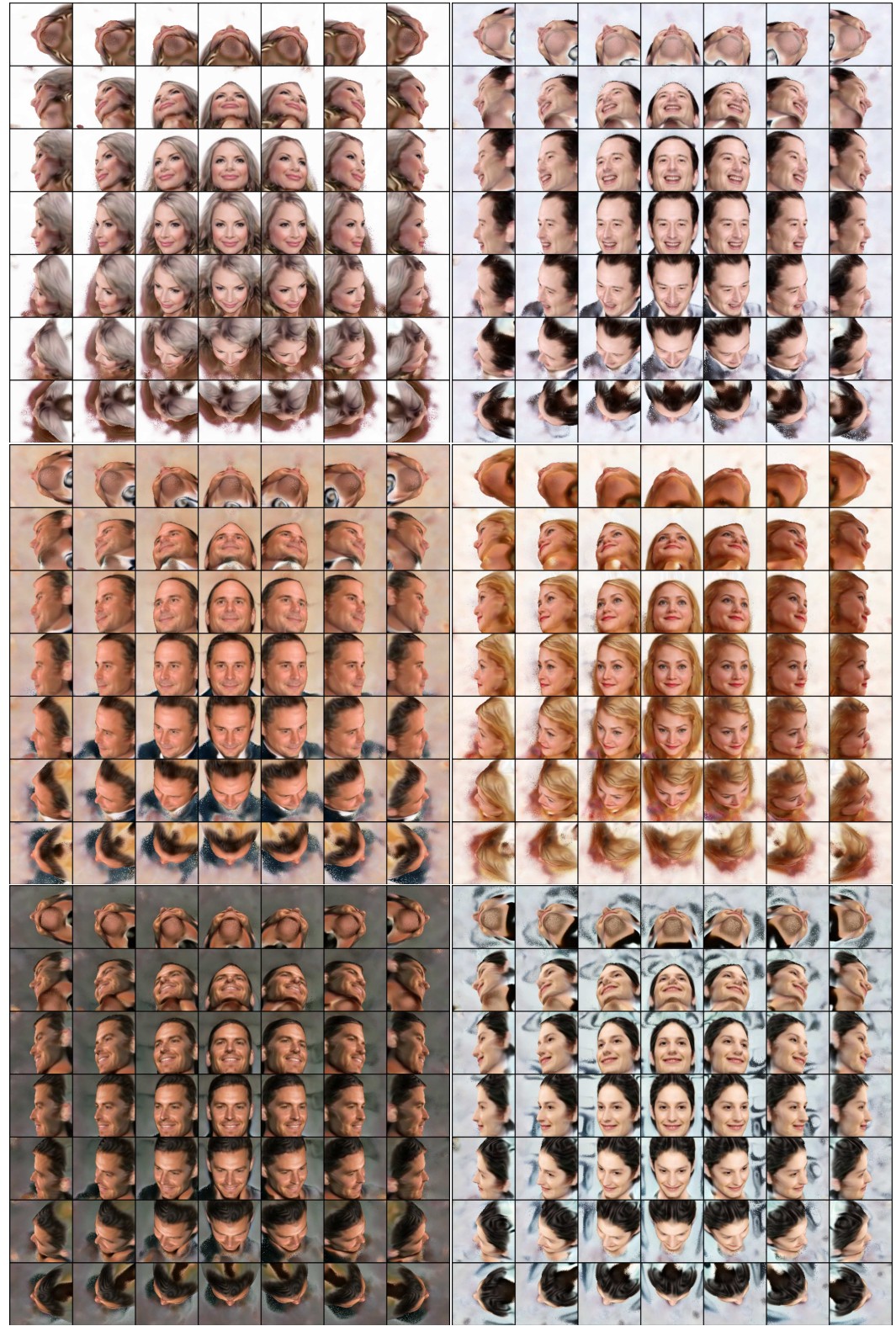

Figure 5: Images generated by our proposed cGOF with different *poses*.

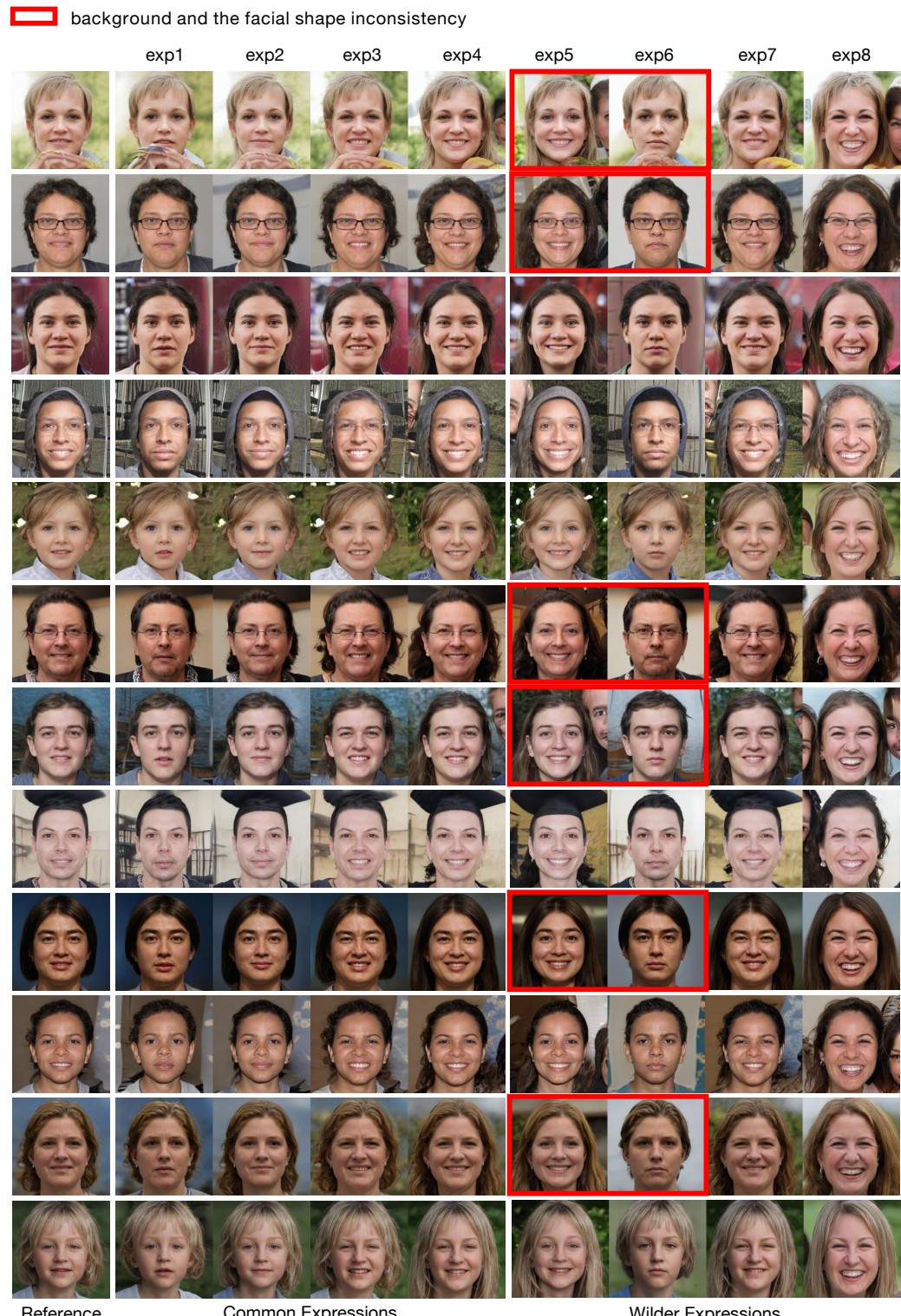

Figure 6: Images generated by GAN-Control [10] with different *expressions*. Each row is generated with the same parameters except the expression code, and each column shares the same expression code. Red boxes highlight examples where GAN-Control produces severe inconsistencies in the identity and background when only the expression is supposed to change. Moreover, we notice the "smiling" attribute tends to strongly correlate with the "female" and "long-hair" attributes.

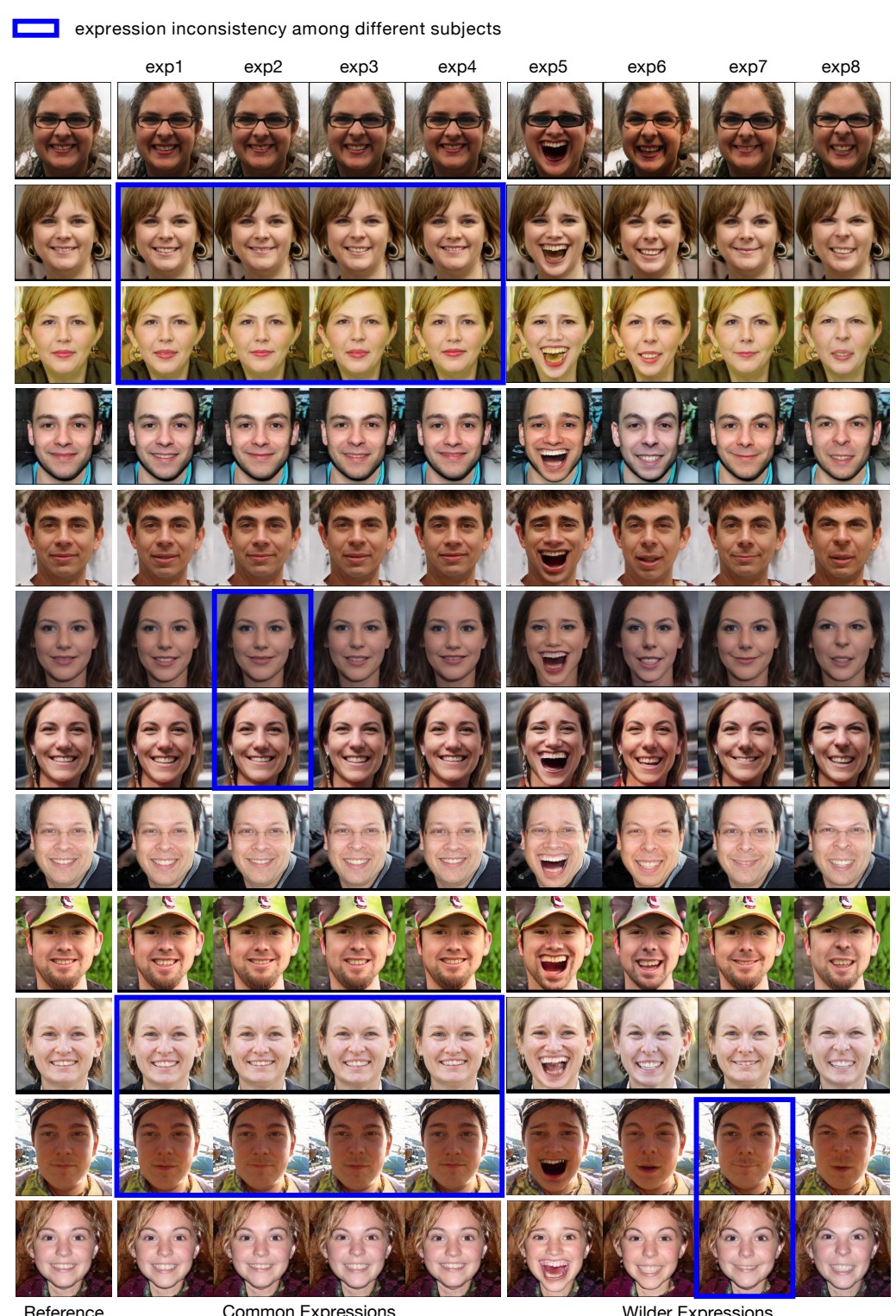

Figure 7: Images generated by DiscoFaceGAN [3] with different *expressions*. Each row is generated with the same parameters except the expression code, and each column shares the same expression code. Blue boxes highlight some examples where DiscoFaceGAN fails to produce consistent expressions among different instances.

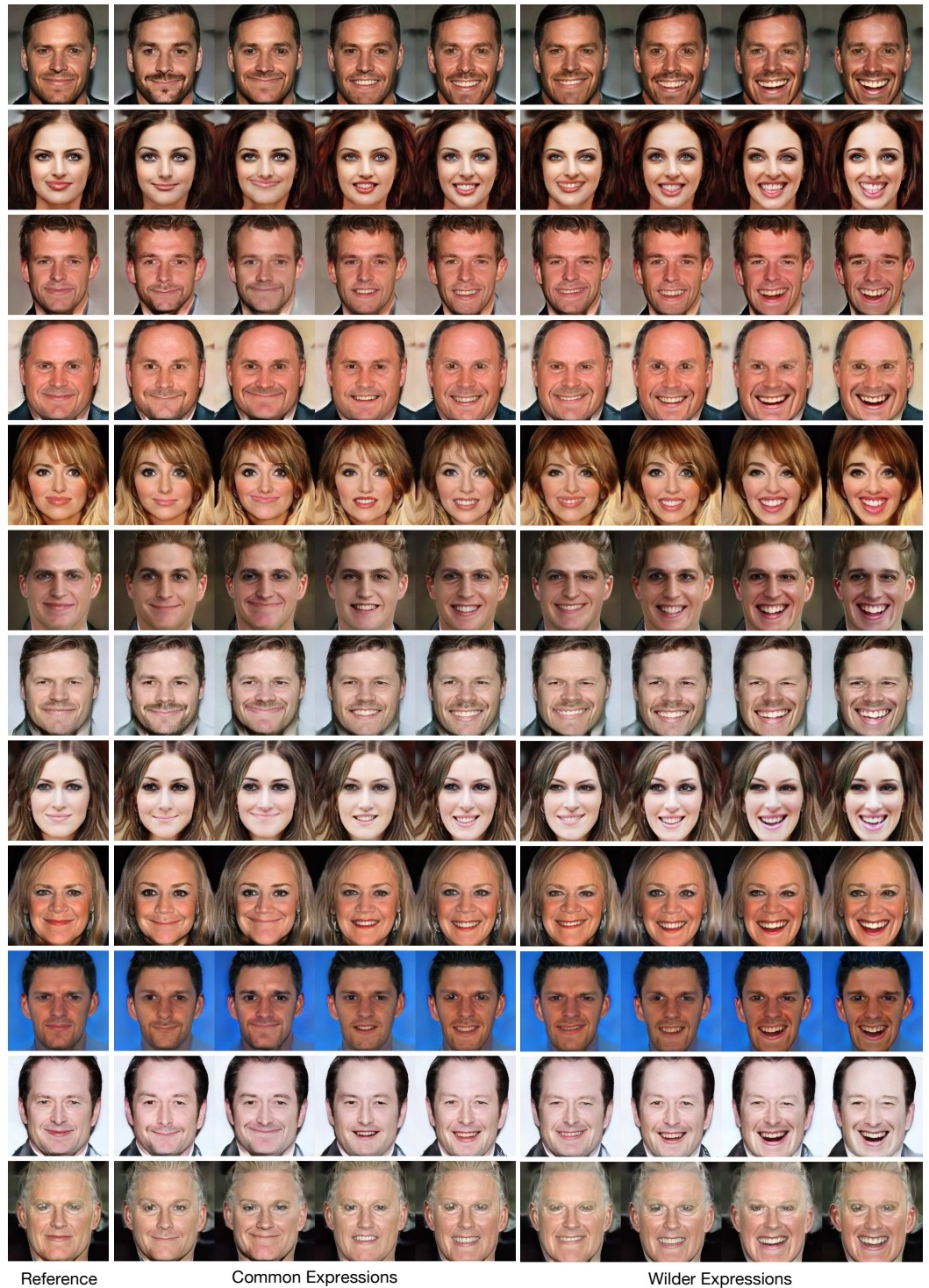

Reference  Common Expressions  Wilder Expressions

Figure 8: Images generated by our proposed cGOF with different *expressions*. Our model generates photo-realistic face images with highly consistent, precise expression control.

exp1   exp2   exp3   exp4   exp5   exp6   exp7   exp8

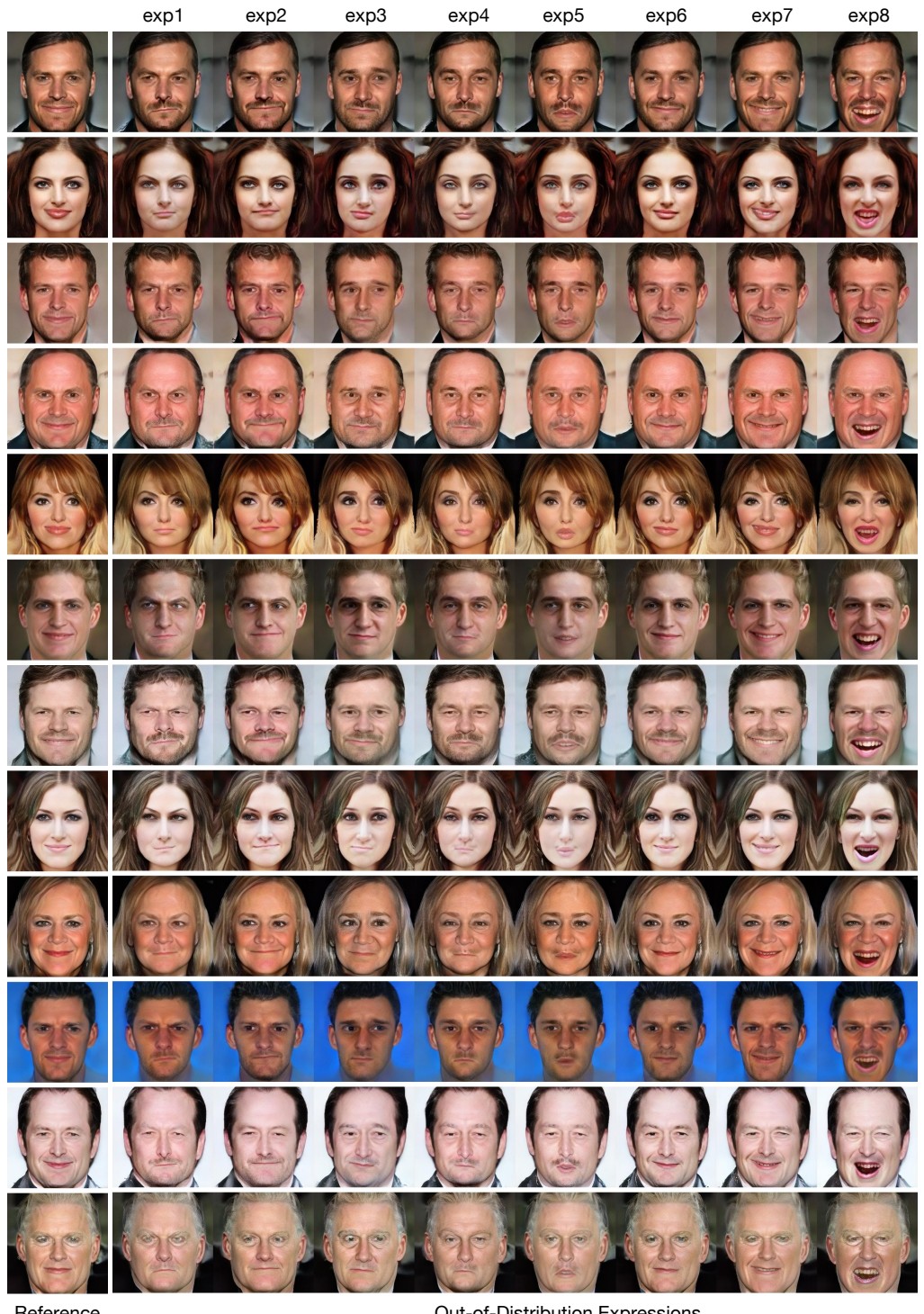

Reference                         Out-of-Distribution Expressions

Figure 9: Images generated by our proposed cGOF with *out-of-distribution expressions*. Our method is capable of synthesizing unseen expressions like raising eyebrow, pouting, curling lips, smirking *etc*.