# OpenReview forum: "Controllable 3D Face Synthesis with Conditional Generative Occupancy Fields"
_NeurIPS.cc/2022/Conference — NeurIPS 2022 Accept_

### Official Review · Reviewer_jXJ9 · 2022-07-09

**Rating:** 5
**Confidence:** 3
**Ethics Flag:** Yes
**Soundness:** 2 fair
**Presentation:** 3 good
**Contribution:** 2 fair

**Summary:**

This paper proposed a new NeRF-based conditional 3D face synthesis framework, which enables the control over head poses and expressions using a parametric 3D face model (3DMM).

The main contribution of this model is a conditional Generative Occupancy Field (cGOF) that effectively enforces the shape of the generated face to commit to the 3DMM model mesh. Besides that, novel loss terms such as 3D landmark loss and volume warping loss are proposed in the system.

The authors trained the model on the CelebA dataset and conducted both quantitative and qualitative comparisons to other methods. The results and comparisons showed that the proposed method outperforms the state-of-the-art methods. Meanwhile, the ablation study was also done to show the effectiveness of each proposed component.


**Questions:**

1. For the 3DMM conditioning, the authors mentioned that they assume the 3DMM coefficients of all the training images follow a normal distribution, while it's not very clear what are the representations of the expression components. Is it also a PCA model or FACs based model? The authors cited [37], which didn't mention about expression components.

2. For the 3D landmark loss, the authors mentioned that they didn't consider the facial contour landmarks due to the impact of occlusion. However, in extreme head poses such as 90 degree left/right view, part of the inner facial landmarks are not visible as well. It's not clear how the authors handle these landmarks and be able to generate plausible results for side view head poses.

3. It would also be great if the authors could include the inference time of the proposed framework.

**Ethics Review Area:**

["Inappropriate Potential Applications & Impact  (e.g., human rights concerns)"]

**Limitations:**

The authors addressed the limitations of the work that the proposed method couldn't handle precise texture and lighting control over the generated face. And also the model relies on 3DMM model, which could not generate high-fidelity shape details such as wrinkles and hair.

For societal impact, the authors mentioned that the work might be used to generate fake facial images. Defensive algorithms on recognizing these images should be developed.

**Strengths And Weaknesses:**

Strengths:

1. This paper proposed a new NeRF-based 3D face synthesis framework, which allows controlling the large head pose change and expression changes of frontal face images.

2. The core contribution is the usage of the conditional Generative Occupancy Field (cGOF) model. Other terms such as 3D landmark loss and volume warping loss terms all contributed to the effectiveness of the proposed framework.

3. This paper is well structured and easy to follow. The authors clearly explained each term and did ablation study to show the impact of each proposed component.

4. The authors further did quantitative and qualitative evaluations on the proposed method and some state-of-the-art methods such as PIE, StyleRig, DiscoFaceGAN, E2E, HeadNeRF, etc, showing that the proposed method outperforms.

Weaknesses:
1. I'm a little concerned that the originality of this proposed work is not enough. For example, the 3DMM conditioning idea is similar to the work in [8] and the cGOF idea is inspired from [53]. It would be great to show that how the way the authors model the 3DMM conditioning in a normal distribution outperforms the VAE model in [8].

2. Some technical details are not very clear to me. Please see the following questions.

---

> ### Author Response · Authors · 2022-08-02
> **Response to Reviewer jXJ9**
>
>
> We appreciate Reviewer jXJ9’s constructive and detailed feedback. We are glad that the reviewer acknowledges the contribution of our proposed conditional Generative Occupancy Field, evaluation and ablation experiments, and well structured writing. We address the concerns and questions below.
>
> **Q: “originality of this proposed work is not enough… 3DMM conditioning idea is similar to the work in [8] and the cGOF idea is inspired from [53]”**\
> **A:** Our baseline model is indeed inspired by DiscoFaceGAN [8] and HeadNeRF [17] leveraging a 3DMM parameter reconstruction loss. However, the baseline results in poor 3D controllability in the generated images as shown in Table 2 in the main paper (row 2).
> To achieve effective 3DMM conditioning, we draw inspiration from GOF [53] and propose a **novel conditional Generative Occupancy Field (cGOF)**.
>
> Notably, unlike GOF [53], where the idea is simply to shrink the volume sample region to a narrow interval through a cumulative rendering process, our proposed **conditional GOF** involves a **Mesh-guided Volume Sampler** that gradually concentrates the samples on a conditional 3DMM mesh. Moreover, we further propose a **Distance-aware Volume Density Regularizer** that effectively suppresses the volume densities away from the conditional mesh.
>
> To the best of our knowledge, this is the first paper to devise a **controllable NeRF-based generative model**. We believe this is a novel and significant attempt towards controllable 3D object synthesis.
>
> **Q: “It would be great to show that how the way the authors model the 3DMM conditioning in a normal distribution outperforms the VAE model in [8]”**\
> **A:**
> Both VAE models in DiscoFaceGAN [8] and the parameter normalization in our paper aim at mapping the normalized gaussian parameters to the 3DMM parameters. Empirically, we found simple normalization achieves good results without the extra complexity of training VAEs as in [8]. In fact, this is not the major difference between DiscoFaceGAN and our proposed method; rather, the key difference lies in the conditioning mechanism, ie., our proposed conditional GOF and the 3D landmark and warping losses.
> Furthermore, we additionally conduct a direct comparison between our method and DiscoFaceGAN [8]. Specifically, we constructed a “Pi-GAN + DiscoFaceGAN” model, by introducing the conditioning methods of DiscoFaceGAN, i.e. imitative loss and contrastive loss, into Pi-GAN. As shown in Table B, the “Pi-GAN + DiscoFaceGAN” model (Row 3) outperforms the Pi-GAN and DiscoFaceGAN baselines, but is still significantly worse than our proposed method. This further demonstrates the effectiveness of our proposed conditioning mechanism. We have added this result to the rebuttal revision paper.
>
> **Table B:  Comparison with the "Pi-GAN + DiscoFaceGAN" model.**
> | Index | Method                | CD       | LD       | LC        | DS_s      | DS_e      | DS_p      |
> |-------|-----------------------|----------|----------|-----------|-----------|-----------|-----------|
> | 1     | Pi-GAN                | 1.09     | 5.04     | 2.04      | 2.13      | 2.54      | 7.16      |
> | 2     | DiscoFaceGAN          | -        | -        | -         | 5.97      | 15.70     | 5.23      |
> | 3     | Pi-GAN + DiscoFaceGAN | 1.32     | 2.40     | 32.55     | 6.16      | 7.63      | 12.13     |
> | 4     | Ours                  | **0.27** | **1.26** | **92.88** | **23.24** | **29.13** | **23.45** |
>
> **Q: "It's not very clear what are the representations of the expression components. Is it also a PCA model or FACs based model? ”**\
> **A:** Following [10], we adopt the widely-used Basel Face Model [37] for shape and texture bases, and use the expression bases of [a], built from FaceWarehouse [b]. The expression components are obtained from a PCA model of the offsets between the expression meshes and the neutral meshes of individual persons. We have added more details in the updated paper.
>
>
> **Q: “It's not clear how the authors handle these (invisible) landmarks.”**\
> **A:** During training, we sample camera poses from a normal distribution, with a vertical standard deviation of 0.15 radian and a horizontal standard deviation of 0.3 radian, roughly estimated from the CelebA dataset. With such pose variations, inner 3D landmarks are in fact hardly occluded. We hence simply assume all landmarks are valid.
>
>
> **Q: “include the inference time”**\
> **A:** With one GTX1080Ti GPU, it takes 0.5716s for cGOF to generate one 128x128 image, and 0.0550s for the neural renderer to upsample it to 512x512. We admit the point-by-point prediction of NeRF-based method is time-consuming. But the optimization of the inference time is not the key objective of this paper, so we leave it to the future work.
>
> **References:**
> - [a] Guo et al. Cnn-based real-time dense face reconstruction with inverse-rendered photo-realistic face images. TPAMI, 2018.
> - [b] Cao et al. Facewarehouse: A 3d facial expression database for visual computing. TVCG, 2013.

---

### Official Review · Reviewer_r2K4 · 2022-07-10

**Rating:** 4
**Confidence:** 4
**Soundness:** 3 good
**Presentation:** 2 fair
**Contribution:** 2 fair

**Summary:**

This paper aims the task of controllable 3D face synthesis based on a set of single-view real-world face images. The authors propose a solution based on the framework of pi-GAN and the widely used parametric 3D Morphable Model (3DMM). The key idea is that the conditional Generative Occupancy Field is employed to explicitly guide the volume sampling procedure of the 3DMM mesh. Meanwhile, for fine-grained 3D shape control, the 3D landmark loss and a volume warping loss is used. In the experiments, both qualitative evaluation and quantitative evaluation show the effectiveness of the proposed method.

**Questions:**

1.  In section 3.2, “where the shape, expression, pose and other factors of a face are modeled by a set of PCA bases and coefficients z = (z_shape, z_exp, z_tex, z_else)”, here, z_tex denotes the factor of pose or texture?

2. Please give the comparsion results of the propsoed method with the baseline model mentioned in the beginning of section 3.2.

**Ethics Review Area:**

["I don’t know"]

**Limitations:**

1. The novolty of this paper is limited.
2. The presentation of the whole paper is not smooth and well structured.
3. The experimental results are not sufficient.

**Strengths And Weaknesses:**

Strengths: The idea of combing pi-GAN and GOF as well as 3DMM for the task of controllable 3D face synthesis shows a certain novelty.

Weaknesses: The novelty of the proposed method is limited. It can not be regarded as a new framework since all the related methods of pi-GAN, GOF and 3DMM are well-known. Meanwhile, the experimental results are not sufficient. The comparison of the proposed method with the Baseline method is missed in Table 1 and Table 2. Moreover, the presentation of the whole paper is not good. For example, in section 3.2, “where the shape, expression, pose and other factors of a face are modeled by a set of PCA bases and coefficients z = (z_shape, z_exp, z_tex, z_else)”, here, z_tex denotes the factor of pose or texture?

---

> ### Author Response · Authors · 2022-08-02
> **Response to Reviewer r2K4**
>
>
> We thank Reviewer r2K4 for the constructive comments. We would like to address a few concerns and questions below.
>
> **Q: “The novelty of the proposed method is limited”**\
> **A:** To the best of our knowledge, this is the first paper to devise a **controllable NeRF-based generative model**. We believe this is a novel and significant attempt towards controllable 3D object synthesis. To achieve 3D controllability, we incorporate priors from a 3DMM face model. However, conditioning a NeRF-based volumetric representation on a mesh-based 3DMM is not trivial at all. In order to achieve effective conditioning, we draw inspiration from GOF and propose a **novel conditional Generative Occupancy Field (cGOF)**.
>
> Notably, unlike GOF, where the idea is simply to shrink the volume sample region to a narrow interval through a cumulative rendering process, our proposed **conditional GOF** involves a **Mesh-guided Volume Sampler** that gradually concentrates the samples on a conditional 3DMM mesh. Moreover, we further propose a **Distance-aware Volume Density Regularizer** that effectively suppresses the volume densities away from the conditional mesh. Our proposed cGOF is not a trivial combination of pi-GAN, GOF and 3DMM, but rather a carefully designed framework for effective 3D controllable NeRF synthesis.
>
> **Q: “Please give the comparison results of the proposed method with the baseline model in section 3.2”**\
> **A:** As explained in Sec 3.2, the baseline model consists of pi-GAN and a 3DMM parameter reconstruction loss $\mathcal{L}_\text{recon}$. The comparison with the baseline is provided in Table 2 in the main paper, of which we copy a part to Table A in the follow for reference. **Row 2** (‘+ $\mathcal{L}_\text{recon}$’) corresponds to the baseline (highlighted in the updated version).
> Compared to the baseline, the full model (**row 9**) achieves significantly better 3D controllability over the generated faces. For example, 3D Landmark Correlation 26.15 -> 92.88, and Disentanglement Scores 3.56 -> 23.24 (shape), 5.03 -> 29.13 (expression), 11.00 -> 23.45 (pose), as shown below.
>
> **Table A:  Ablation Study extracted from Table 2 in the main paper.**
> | Index | Method                       | CD       | LD       | LC        | DS_s      | DS_e      | DS_p      |
> |-------|------------------------------|----------|----------|-----------|-----------|-----------|-----------|
> | 2     | + $\mathcal{L}_\text{recon}$ | 0.87     | 3.85     | 26.15     | 3.56      | 5.03      | 11.00     |
> | 9     | Ours                         | **0.27** | **1.26** | **92.88** | **23.24** | **29.13** | **23.45** |
>
> **Q: Typo**\
> **A:** Thanks for pointing it out. $z_\text{tex}$ denotes the component of the code that controls the texture, not the pose. We have fixed it in the updated version.

---

> ### Comment · Reviewer_r2K4 · 2022-08-07
> **Change my Rating as Accept**
>
> All my concerns have been feedback by the authors. According to the authors' response and other reviewers' comments, I change my Rating of this manuscript as Accept.

---

> > ### Author Response · Authors · 2022-08-09
> > **Thank you**
> >
> > We are glad that your concerns have been addressed. Thank you for your valuable comments and for taking the time to respond to the rebuttal.

---

### Official Review · Reviewer_PM1k · 2022-08-02

**Rating:** 7
**Confidence:** 3
**Soundness:** 4 excellent
**Presentation:** 4 excellent
**Contribution:** 3 good

**Summary:**

The paper attacks the standard but hard problem of starting with a single-view 2D face shot and creating a 3D face model that can take expression and pose as inputs. Their approach is to meld pi-GAN with 3D Morphable Modeling (3DMM) of priors (based on PCA basis faces) and Generative Occupancy Field (GOF).  I believe that they are also proposing the 3D Landmark Loss for this application, and a Volume Warping Loss which is a nice conservation type prior.

**Questions:**

-  "we assume that the 3DMM coefficients of all the training images follow a normal distribution".  Surprised that you don't need a conditional covariance matrix for that, does it just not matter much?

- I may have missed it, but I didn't see where or how you chose the different values for lambda's in (9)?

- I did not understand this sentence "This allows us to incorporate a 3DMM parameter reconstruction loss that ensures the
generated face images to commit to the input condition."

**Limitations:**

Authors note the obvious problem in this field of creating fakes. In fact, it is hard to conceive of when you would want to do this other than trying to fake people out, but while I think this is not a really important problem to work on, I recognize that in general 3D from 2D is a popular set of problems and do not think the paper should be rejected for its choice of problem.

**Strengths And Weaknesses:**

Strengths:
- The proposals seem totally reasonable: they bring together solid components and good ideas.
- The comparisons to state-of-the-art seem fine, and I appreciated the additional results in the Supplemental
- They show large gains to controllability
- The ablation study was useful to see
- I found the paper easy to read and mostly well-written

Weaknesses:
- This is pretty close to HeadNeRF but with the extreme case of just 1 input view, I would have appreciated more discussion of that
- While it's nice their loss function in (9) balances many goals, they end up with 6 (5 free) hyperparameters lambda to set
- Metrics: it would be nice to see a really proper human rater paired comparisons survey done (but yes I see the qualitative results in the Supplemental)
- They limit the generated images to 64 x 64 during training (and then use another Neural Renderer to map it to 512 x 512... this seemed a bit iffy), and then Table 1 suggests their method without the extra Neural Renderer step). I am not sure I grokked the full ramifications and comparison issues due to this.

---

> ### Author Response · Authors · 2022-08-07
> **Response to Reviewer PM1k**
>
> We appreciate Reviewer PM1k's constructive and detailed feedback. We are glad that the reviewer acknowledges the contribution of our proposed method and likes our results. Below we address the concerns and questions.
>
> **Q: "This is pretty close to HeadNeRF"** \
> **A:**
> First, our method is a generative model trained via a GAN loss, whereas HeadNeRF is trained via an image reconstruction loss with precomputed 3DMM parameters as input. As a result, our model produces much higher fidelity images than HeadNeRF, as shown in the supplementary material.
>
> Moreover, HeadNeRF enforces 3DMM conditioning simply via a reconstruction loss. We have demonstrated in our experiments that parameter reconstruction is an indirect conditioning approach and is ineffective for precise controllability, as shown in row 2 of Table 2 “$+ \mathcal{L}_\text{recon}$” (baseline). Our key technical contribution is **precisely a new effective conditioning mechanism** via a novel conditional Generative Occupancy Field (cGOF). It effectively conditions the NeRF on a given 3DMM mesh through a novel **Mesh-guided Volume Sampler** and a **Distance-aware Volume Density Regularizer**. Together with two losses (3D landmark and volume warping), our proposed method achieves significantly better 3D controllability, as shown in Table 1 & 2 and the supplementary material.
>
> **Q: "I didn't see where or how you chose the different values for lambda's in (9)"** \
> **A:**
> We start from the original Pi-GAN and add the proposed components one by one, as in Table 2 in the main paper. For each component, we first initialize the hyperparameter lambda as $1$, and empirically find a reasonable value before taking a fine-grained search. We have added these details to the updated version.
>
> **Q: "Metrics: it would be nice to see a really proper human rater paired comparisons survey done"** \
> **A:**
> To address this, we conduct a user study to add to the comparison. We follow the experiment setting of Table 1 in the main paper. Specifically, we provide the control results of five methods, and ask a total of $21$ users to rank the results according to their image quality and the controlling effects.
>
> Table C summarizes the results, where the "Average Ranking / Average Score / Ranking 1st Ratio" are reported for each method with respect to concerning factors. Our model achieves the highest average ranking and scores for all aspects, indicating our model produces more perceptually compelling results and achieves better 3D controllability than other counterparts. We have added the user study to the updated version.
>
> **Table C. User Study on the Disentangle Performance and Image Quality. Each cell contains "Average Ranking / Average Score / Ranking 1st Ratio".**
> |               |      DiscoFaceGAN    |     GAN-Control     | Pi-GAN + $\mathcal{L}_\text{recon}$ |     HeadNeRF      |       cGOF (Ours)       |
> |:-------------:|:--------------------:|:-------------------:|:-----------------------------------:|:-----------------:|:-----------------------:|
> |    Identity   |  3.05 / 59.1 / 9.52% | 2.57 / 68.6 / 0.00% |3.33 / 53.3 / 9.52%| 4.67 / 26.7 / 0.0% | **1.38 / 92.4 / 81.0%**|
> |   Expression  |  2.29 / 74.3 / 19.1% | 3.81 / 43.8 / 9.52% |3.19 / 56.2 / 0.00%| 4.29 / 34.4 / 0.0% | **1.43 / 91.4 / 71.4%** |
> |      Pose     |  2.57 / 68.6 / 14.3% | 3.71 / 45.7 / 0.00% |3.81 / 43.8 / 4.76%| 3.67 / 46.7 / 0.0% | **1.24 / 95.3 / 81.0%**|
> | Image Quality |  2.10 / 78.1 / 9.52% | 2.86 / 62.9 / 4.76% |3.86 / 42.9 / 0.00%| 5.00 / 20.0 / 0.0% | **1.19 / 96.2 / 85.7%** |
>
> **Q: "They limit the generated images to 64 x 64 during training. I am not sure I grokked the full ramifications and comparison issues due to this."** \
> **A:** Due to the GPU memory limit, we train the model with 64 x 64 images. During inference, we render 128 x 128 images. With the super-resolution module [50], we are able to enhance the image quality and generate 512 x 512 images, without much damage to the controllability.
>
> **Q: "Surprised that you don't need a conditional covariance matrix for that, does it just not matter much."** \
> **A:**
> Thanks for pointing out this mistake. In fact, we did calculate the covariance matrix and use it to normalize and denormalize the 3DMM coefficients, as shown in the submitted code (see "parse" function in the "parse_recon.py" and "norm_coeff" and "denorm_coeff" in the "facerecon_model.py"). We did encounter some problems when simply using the standard deviation in the beginning, but forgot to update the writing. We have fixed it in the updated version.
>
> **Q: "I did not understand this sentence 'This allows us to incorporate a 3DMM parameter reconstruction loss that ensures the generated face images to commit to the input condition.'"** \
> **A:**
> Essentially, the baseline exploits a 3DMM reconstruction model that predicts 3DMM parameters from the generated images. We then enforce the predicted 3DMM parameters to be close to the ones given as input conditions to generate the images.

---

### Review · Ethics_Reviewer_6uVt · 2022-08-05

**Recommendation:**

No recommendation. The authors already acknowledge potential inappropriate applications in the paper.

**Ethical Issues:**

Yes

**Ethics Review:**

As both reviewers and authors note, there are Inappropriate Potential Applications & Impact.

---

### Meta-Review · Area_Chair_HY3F · 2022-09-04

**Recommendation:** Accept
**Confidence:** Certain

**Metareview:**

Paper attacks a hard problem and brings together state-of-the-art ideas to demonstrate substantial wins.  Many good points were raised by the reviewers, and we ask the authors to carefully read through the feedback and address what they can for the final version.

**Award:**

No

---

### Decision · Program_Chairs · 2022-09-14

Accept